# Agricultural Cultivation Structure in Arid Areas Based on Water–Carbon Nexus—Taking the Middle Reaches of the Heihe River as an Example

**Boxuan Li [1], Meng Niu [2,\*], Jing Zhao [3], Xi Zheng [3], Ran Chen [3], Xiao Ling [3], Jinxin Li [3] and Yuxiao Wang [1]**

[1] Faculty of Architecture, Civil and Transportation Engineering, Beijing University of Technology, Beijing 100124, China; liboxuan@emails.bjut.edu.cn (B.L.); wangyuxiao@emails.bjut.edu.cn (Y.W.)

[2] China Urban Construction Design and Research Institute, Beijing 100120, China

[3] School of Landscape Architecture, Beijing Forestry University, Beijing 100083, China; zhaojing@bjfu.edu.cn (J.Z.); zhengxi@bjfu.edu.cn (X.Z.); chenran705367787@bjfu.edu.cn (R.C.); lingxiao@bjfu.edu.cn (X.L.); tbts1458@edu.com (J.L.)

\* Correspondence: niumeng@cucd.cn

**Abstract:** China faces challenges of food security and sustainable agricultural production. However, current studies rarely address the spatial distribution patterns of water consumption and carbon emissions. We studied the irrigation water use efficiency and carbon emission differences of crops in arid areas and their spatial distribution using wheat and maize, two major food crops in the middle reaches of the Heihe River, as examples. Furthermore, we have optimized low-carbon cropping of crops under the multiple objectives of water conservation and economic development. The results show that: (1) The carbon emissions per unit of water consumption for maize are $0.03 \times 10^{-6}$ t mm$^{-1}$ and $0.49 \times 10^{-6}$ t mm$^{-1}$ for wheat. Irrigation water consumption per unit yield is 515.6 mm t$^{-1}$ for maize and 426.7 mm t$^{-1}$ for wheat. (2) The spatial distribution patterns of irrigation water consumption were opposites for maize and wheat. The former has lower irrigation water consumption in the planting area upstream of the Heihe River and higher in the lower reaches. In contrast, the pattern of wheat irrigation is the opposite. (3) After optimizing the cropping mix for both crops, the area planted with wheat should be reduced to 59% of the current size, while maize should be expanded to 104%. The results of the research hold immense importance in guiding the future grain crop planting patterns for water-saving agriculture and low-carbon agriculture development in arid zones worldwide, aligning with the United Nations' Sustainable Development Goals.

**Keywords:** water–carbon nexus; farming structure; irrigation system; low-carbon agriculture; arid regions; middle reaches of the Heihe River

## 1. Introduction

As the world's population grows exponentially, food security has become a worldwide issue [1,2]. China is currently confronted with the dual challenges of food security and achieving ecologically sustainable development [3]. This poses a challenge to China's agricultural production, especially the irrigated agriculture that guarantees a stable supply of food [4,5]. Despite having only 6% of the world's fresh water resources and 9% of its arable land, China's irrigated agriculture plays a pivotal role in supporting 21% of the global population [6]. Current agricultural production faces pressure on water resources due to the uneven spatial distribution of resources [7,8]. Furthermore, crops exhibit varying responses to different climatic conditions. This leads to an unoptimized agricultural production structure, which in turn results in substantial additional water consumption and greenhouse gas emissions [9,10]. These phenomena are particularly evident in arid regions with fragile ecosystems and high environmental variability [2,11]. For arid zones, water consumption for agricultural production may account for 90% of total regional

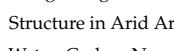



water consumption [12,13]. Therefore, finding ways to maintain agricultural production and ensure an adequate supply of food crops, while simultaneously minimizing water consumption and greenhouse gas emissions, has become an urgent issue that needs to be addressed.

Agricultural carbon emissions are influenced by a variety of factors, of which cropping structure is one of the main ones [14,15]. Various scholars have proposed different possibilities for optimizing the crop cultivation structure. Scholars studying carbon and water footprints across various crops primarily lead research on water use efficiency and carbon emissions [16–18]. For example, Feng et al. measured the water footprint of global crop production and found that the world water footprints of maize, wheat, and rice were 0.73 m$^3$ kg, 1.136 m$^3$ kg, and 1.269 m$^3$ kg, respectively [19]. This is a preliminary indication of the differences in irrigation water consumption between crops. In contrast, Mekonnen and Hoekstra measured the water footprint of wheat as essentially 114% of that of maize for the period 1996–2005 [20]. The concept of footprints can examine the growth patterns of various crops on a comprehensive scale. However, they are primarily designed for large-scale assessments, such as at the country level, and may not adequately guide specific regional agricultural cultivation practices. At the same time, the data in various studies are obtained from satellite data for a particular crop sum, lacking the exploration of the pattern of geographical distribution of crops. To guide agricultural cropping patterns more precisely and directly, it is particularly important to construct an irrigation model with higher accuracy and that is more specific to the production life of farmers. In another study, an agricultural structural adjustment model with computer simulation obtained a 34.35% reduction in the area planted with food crops in the optimal agricultural irrigation water-saving model [21]. This demonstrates the current urgency of renewing the cropping structure of food crops if the degree of water saving in regional agricultural irrigation is to be regulated from a macro perspective.

Arid zones have unique climatic conditions and crop cultivation structures. It is important to select research crops and research models for their special characteristics. Arid regions possess distinct geographical conditions, and climate change significantly impacts the water cycle within these areas [22,23]. Previous studies have proven that the scale of major food crops such as maize will drop by about 5% after water-saving irrigation is applied to agriculture, while the scale of bean and oil crops will increase significantly [24]. In the current context of food security, we take food crops as the main object of research. Previous studies have shown that adjusting crop planting structures according to irrigation water use efficiency can greatly improve agricultural water use efficiency [25]. For irrigated agriculture, the appropriate increase in the frequency of planting can effectively reduce the water consumption of crops while ensuring crop yields when the appropriate season, location, and other factors are suitable for crop growth [26].

In summary, the current research on agricultural planting structure primarily focuses on a national or regional level. In order to protect the interests of farmers and provide practical guidance for agricultural production, there is a need to construct a higher-precision model that can accurately reflect the spatial heterogeneity of water resource consumption and carbon emissions during irrigation processes. Furthermore, in order to better guide low-carbon agricultural production, particularly the adjustment of grain production scale, it is necessary to optimize the agricultural planting structure.

Therefore, this research holds two significant values. Firstly, we have developed a precise water–carbon nexus model for agricultural irrigation processes at the county level. This model accurately delineates the spatial variations in carbon emissions and irrigation water consumption between two high-value crops. It serves as an effective guide for optimizing agricultural production practices. Secondly, we have devised an updated planting structure that integrates considerations of food security, farmers' income, and the reduction of irrigation water consumption and carbon emissions in agricultural production. This research aims to establish the spatial distribution data of major food crops, namely maize and wheat, in the middle reaches of the Heihe River with town-level accuracy. This

is accomplished by employing the water–carbon nexus framework in a typical arid zone. In the current global context, there is a pressing need for concerted efforts to mitigate the adverse impact of agriculture on the climate and ensure food production. Under the guidance of the United Nations Sustainable Development Goals (SDGs), the pursuits of a secure water environment, low-carbon emission environmental protection, sustainable agricultural development, and food security are globally unified objectives. Our findings offer valuable guidance for the cultivation of food crops in arid regions worldwide, facilitate the exploration of spatial distribution patterns of water consumption and carbon emissions among different crops, and offer valuable insights towards achieving an optimal crop cultivation pattern that ensures food security.

## 2. Materials and Methods

### 2.1. Study Area

In order to construct a typical arid zone, the middle reaches of the Heihe River were selected as the main research object in this study. The Heihe River basin is the second largest inland river basin in northwest China, with a total length of about 1100 km and a basin area of about 130,000 km$^2$. As a typical arid zone, the Heihe River basin receives little precipitation, and crop cultivation depends mainly on surface water and groundwater supply. In severe cases, agriculture will consume 96% of the region's water [27]. In recent decades, the Heihe River has experienced a dramatic decrease in water volume, a reduction in the number of tributaries in the middle and lower reaches, and a significant reduction in flow, with serious ecological changes [28]. There is an urgent need for the region to address the issue of high water consumption in agriculture and ecological sustainability.

Zhangye is situated in the middle reaches of the Heihe River and is recognized as one of China's most agriculturally developed regions. The arable land in Zhangye accounts for 95% of the Heihe River basin, as depicted in Figure 1. However, over the past decade, the middle reaches of the Heihe River have witnessed an expansion of oases aimed at enhancing the scope of agricultural cultivation. Approximately 100,000 hectares of irrigated farmland have been reclaimed, leading to a fragmented arable landscape within the intricate topography of the middle Heihe River region. Consequently, this has resulted in a lack of scalability in crop cultivation [29,30].

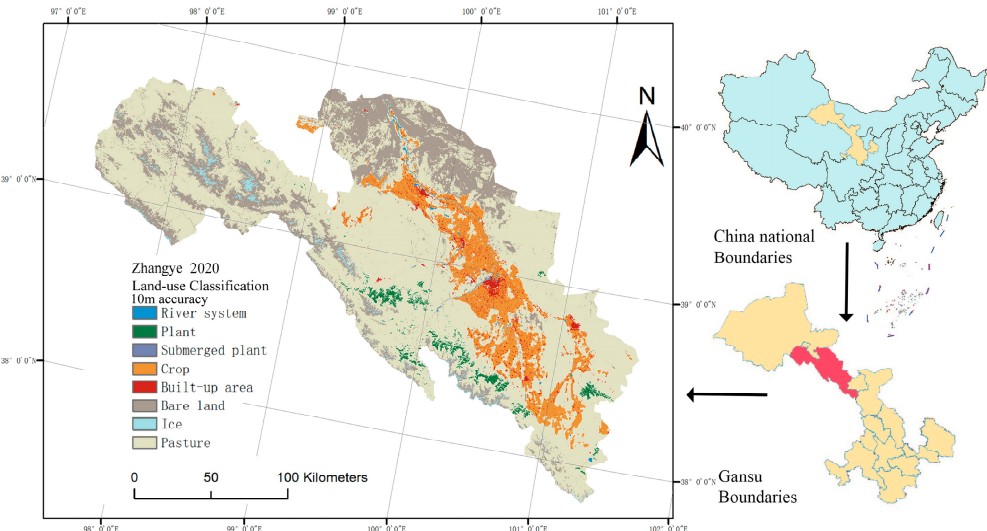

**Figure 1.** Zhangye City Location and Land Use Data.

### 2.2. Methods and Data

In this study, the planting distribution and yield distribution of two crops in the middle reaches of the Heihe River were determined using Landsat and MOD13 satellite data, along with digital terrain model (DEM) elevation data and land use data provided by

the China Geospatial Data Cloud. Finally, the accuracy of the results was validated using agricultural data from China and Zhangye City Statistical Bureau.

In this study, crop evapotranspiration and effective precipitation were calculated using meteorological data to obtain irrigation water consumption data for the two crops. Additionally, the carbon emission data of the irrigation process for these crops were obtained based on farmers' survey data and carbon emission coefficients.

Finally, the study incorporated a linear programming model that integrates factors such as food supply and farmers' income to derive the optimal low-carbon crop cultivation model. The detailed flow chart is shown in Figure 2.

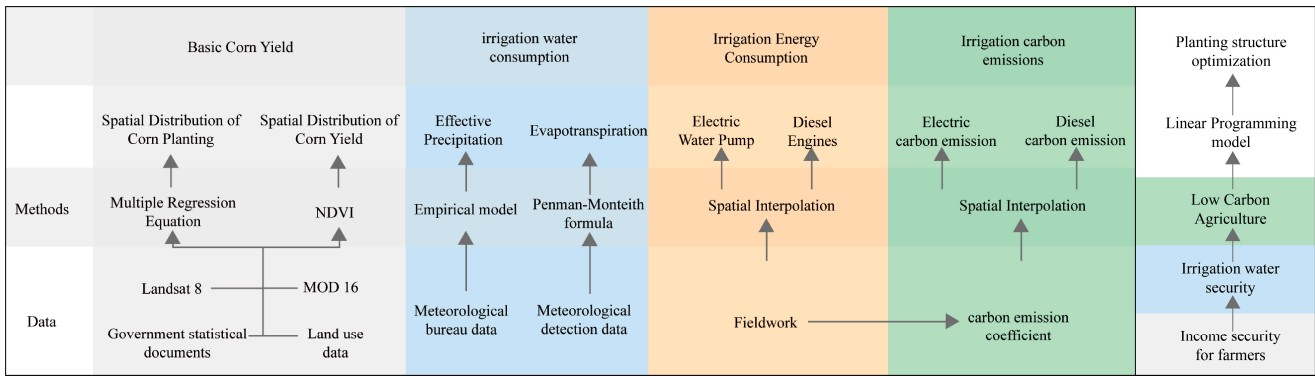

**Figure 2.** Research framework diagram.

### 2.2.1. Data Source

The detailed sources of data for this study are as follows: (1) Daily precipitation data during the crop-growing season, including measurements of daily precipitation, relative humidity, maximum and minimum temperatures, sunshine hours, and average wind speed, were primarily obtained from the National Meteorological Information Center (CMA), available at http://data.cma.cn/. (2) Satellite data, such as Landsat 8 satellite data and DEM elevation data, were predominantly sourced from the Geospatial Data Cloud at http://www.gscloud.cn/. Additionally, MOD16 data were acquired from the NTSG website accessible at http://www.ntsg.umt.edu/. (3) Land use data were extracted from Esri's global 10 m resolution land use data, which can be accessed at https://www.arcgis.com/apps/instant/media/index/. (4) Carbon emission coefficients for electricity and diesel were obtained from the Ministry of Ecology and Environment of the People's Republic of China, and the relevant information is available at https://www.mee.gov.cn/. (5) Statistics on grain production and arable land area primarily came from the Zhangye Statistical Yearbook. However, since the official statistical yearbook for 2022 has not been released by Zhangye, some data were derived from an analysis of the data from the past five years. (6) Survey data were mainly derived from individual interviews conducted with local farmers. In order to demonstrate the significant influence of topographic factors and the distribution of irrigation water resources on the irrigation process, this study specifically selected four representative villages in the middle reaches of the Heihe River basin, where maize and wheat cultivation is predominant. Each farming household cultivates a non-separable area of 1–2 hectares for both crops. The proportional distribution of major cereal crop cultivation areas in each county is presented in Table 1. These villages are situated in diverse geographical and topographical settings. The distribution of questionnaires took place in February 2023, resulting in a collection of 421 completed questionnaires. Among them, 409 questionnaires were deemed qualified, while 12 were considered unqualified, yielding a qualification rate of 97.1%.

**Table 1.** The proportional distribution of major cereal crop cultivation areas.

| Point Number | Wheat | Corn | Tubers |
|---|---|---|---|
| Zhangye City (Total) | 0.17 | 0.48 | 0.10 |
| Ganzhou District | 0.04 | 0.92 | 0.00 |
| Sunan County | 0.14 | 0.27 | 0.03 |
| Minle County | 0.38 | 0.13 | 0.31 |
| Linze County | 0.01 | 0.97 | 0.00 |
| Gaotai County | 0.10 | 0.65 | 0.04 |
| Shandan County | 0.42 | 0.03 | 0.23 |

### 2.2.2. Data Integration at Different Scales

In this study, we used different sources of data with different precision. For example, satellite data are from open-source websites and their data type is raster; meteorological data are attribute data with precision to different counties; survey data are point data. In order to obtain geographic data with precision to different counties, we used various spatial processing techniques, such as Euclidean distance, spatial algorithms, etc. The processing of different data is shown in Figure 3.

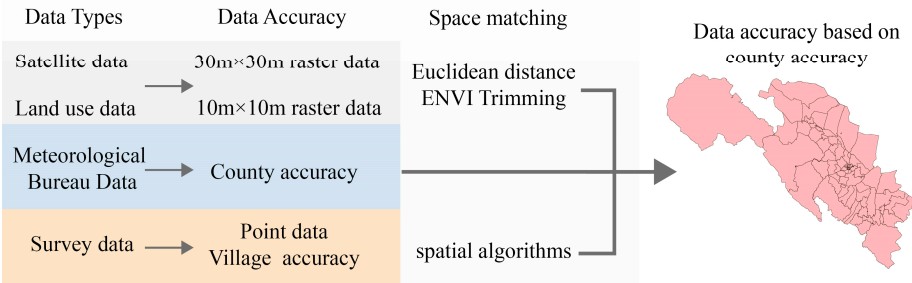

**Figure 3.** Flowchart of data integration at different scales.

### 2.2.3. Constructing Carbon Emission Linkage of Irrigated Waters for Agricultural Irrigation Process

There are many aspects of agricultural production processes that involve greenhouse gas emissions. In order to better construct the relationship between irrigation water consumption and carbon emissions, the irrigation process in agricultural production is selected in this study [31]. The irrigation process, as an important link in agricultural production affecting the water–carbon nexus, involves numerous energy conversion links [32]. In this paper, we focus solely on the agricultural irrigation process, specifically examining the mechanical aspects of water extraction, storage, and delivery. By narrowing our scope to these elements, we aim to establish a more direct correlation between irrigation water consumption and carbon emissions [33].

### 2.2.4. Sown Area and Yield

To enhance the accuracy of crop distribution data, remote sensing data was employed to estimate the crop cultivation distribution in the region. Subsequently, the obtained data was cross-validated with the official records to ensure its reliability and accuracy. To obtain a more accurate spatial distribution of the two major food crops, satellite image data were utilized. The processing of satellite image data involves two main steps: Preprocessing and crop remote sensing identification.

To begin with, Landsat 8 data were employed to obtain the normalized difference vegetation index (NDVI) for the region. Through radiometric correction, atmospheric correction, and image cropping using the ENVI software (Version: 5.6), the precise NDVI temporal curve of the region was derived. ENVI software is developed by Exelis Visual Information Solutions, and the main purpose of this software is to process and analyze

remote sensing data. The NDVI values were calculated for each year and time period, resulting in the generation of NDVI change curves for each year.

To ensure the accuracy of the planting area distribution data obtained from satellite imagery in this study, it was necessary to conduct a thorough verification. Several evaluation methods were employed to validate the NDVI data, including kappa coefficients, confusion matrix analysis, squared mean error, and normalized squared mean error. These techniques were chosen to assess the reliability and precision of the obtained data. After years of verification using NDVI data, the overall classification accuracy of Landsat 8 data for the planting area of the two crops we measured, employing segmented multispectral analysis, has been found to approach approximately 90%.

By comparing the similarity between the NDVI curves of different years and the reference curve, the NDVI values were efficiently normalized using the Euclidean distance. Furthermore, the crop distribution information was obtained by performing weighted calculations based on pixel data for each year. On this basis, specific NDVI curves for two crop growth cycles were used to simulate the yields of maize and wheat. The NDVI curves for crops at different growth stages followed parabolic trends and simulated crop yields in the area.

### 2.2.5. Measurement of Water Consumption in Agricultural Irrigation

To determine the irrigation water consumption (IWC), the actual irrigation water consumption of two crops was measured using Meteorological Bureau data and meteorological detection data. A comprehensive calculation process was employed, involving three main steps. The first step involves calculating evapotranspiration ($ET_c$) and effective precipitation (EP) using meteorological data and information on the area dedicated to maize planting.

In the second step, the IWC is derived by subtracting the EP from the $ET_c$. This calculation provides an estimation of the actual water consumption required for irrigation.

Finally, the spatial distribution of IWC is obtained using the inverse distance weighting method. This technique considers the proximity of data points to determine the water consumption pattern across the designated area.

Crop evapotranspiration ($ET_c$) represents the water evaporated from the ground during crop growth due to photosynthesis and respiration. It directly indicates the amount of unused water in the crop growth process. To calculate $ET_c$, potential evapotranspiration ($ET_p$) is first estimated based on meteorological data and then multiplied by the crop coefficient.

In this study, the Penman–Monteith equation was utilized to calculate $ET_p$. This equation approximates net evaporation using meteorological data as a substitute for direct measurement. Widely adopted, the equation is adopted by the Food and Agriculture Organization of the United Nations for simulating potential evapotranspiration [34,35].

The crop coefficient is an empirical value that accounts for variations between specific crops and reference evapotranspiration standards. It serves as a multiplier to adjust $ET_p$ for different crop types. The calculation equation for $ET_c$ is as follows:

$$ET_c = K_c \times ET_0 \tag{1}$$

where $K$ is the crop coefficient for maize and wheat; $ET_c$ is the crop evapotranspiration (mm/day); $ET_0$ is the potential evapotranspiration (mm/day).

The estimation of crop evapotranspiration ($ET_c$) for specific crops, accounting for their unique characteristics and evapotranspiration standards, can be achieved by multiplying the reference evapotranspiration ($ET_p$) by the crop coefficient.

Effective precipitation (EP) refers to the portion of precipitation that successfully infiltrates the soil and is absorbed and utilized by crops. Empirical models are commonly used to calculate EP based on meteorological precipitation data. In this study, the calculation model utilized was the United States Department of Agriculture Soil Conservation Service (USDA SCS) model [36]. The specific formula is as follows:

$$EP = \begin{cases} \frac{R(4.17-0.2R)}{4.17}, R < 8.3mm/d \\ 4.17 - 0.1R, R \geq 8.3mm/d \end{cases} \tag{2}$$

where $R$ represents daily precipitation.

Ultimately, the computation for determining the irrigation water consumption (IWC) is outlined as follows:

$$IWC = Ep - Et_c \tag{3}$$

where *IWC* is the actual irrigation water consumption of both crops. Ultimately, the spatial allocation of irrigation water consumption is derived utilizing the inverse distance weighting method.

### 2.2.6. Carbon Emissions

Since this study focused on the irrigation process and specifically targeted carbon emissions resulting from machinery usage during irrigation, we were able to accurately collect carbon emission data at the farmer scale. In order to obtain energy consumption data for a typical village, we conducted visits to farmers and recorded the electricity and diesel consumption of pumping machines used by different farmers for cultivating the two selected crops. By employing the spatial interpolation method, we extrapolated the comprehensive energy consumption data for the middle reaches of the Heihe River across the agricultural areas dedicated to the two crops. Carbon emission data were derived from the energy consumption data using the emission factor method. The specific measurement process is as follows:

$$EI_{\text{village},\alpha} = \frac{\sum\limits_{\alpha=1}^{n} E_{village,\alpha}}{\sum\limits_{\alpha=1}^{n} A_{village,\alpha}} \tag{4}$$

$$CI_{\text{village},\alpha} = \frac{C_{village,\alpha}}{A_{village,\alpha}} = \frac{\varepsilon \sum\limits_{j=1}^{n} E_{village,j}}{A_{village}} \tag{5}$$

$$ECC = \frac{\sum\limits_{\alpha=1}^{n} EI_{village,\alpha}}{n} \tag{6}$$

$$CEC = \frac{\sum\limits_{\alpha=1}^{n} CI_{village,\alpha}}{n} \tag{7}$$

where $EI_{\text{village},\alpha}$ represents energy consumption per unit area of $\alpha$ village; $E_{village,\alpha}$ represents irrigation energy consumption of farmers in $\alpha$ village; $A_{village,\alpha}$ represents maize planting area of farmers in $\alpha$ village; $CI_{\text{village},\alpha}$ represents the carbon emission per unit area of $\alpha$ village; $C_{village,\alpha}$ represents total carbon emissions of $\alpha$ village; $\varepsilon$ represents the carbon emission factor. ECC represents the energy consumption per unit area, while CEC represents the carbon emissions per unit area.

The second step entails employing ArcGIS to establish a data model for regional energy consumption, utilizing the spatial distribution of planting areas as a foundation. ArcGIS software (Version: 10.6) was developed by the Environmental Systems Research Institute (Esri) to manage, analyze, and visualize geographic data. Spatial algorithms, a widely utilized technique in geographic information science, are employed to create continuous data on regional grids by extrapolating from existing discrete spatial data. Subsequently, a database of regional irrigation energy consumption and carbon emission coefficients is generated based on this step. In order to minimize calculation errors, the constructed database undergoes meticulous numerical calibration to rectify any abnormal values. Lastly,

the energy consumption and carbon emission coefficients of different villages and towns are horizontally compared using the database data.

$$E_{township} = EI \times A_{township} \tag{8}$$

$$C_{township} = CI \times A_{township} \tag{9}$$

where $E_{township}$ represents the cumulative energy consumption associated with irrigation activities within townships.; $C_{township}$ represents the aggregate carbon emissions attributed to townships; $A_{township}$ represents the total planting area.

### 2.2.7. Planting Structure Optimization Model

A linear programming model was used to optimize the grain crop cultivation pattern in the middle reaches of the Heihe River. The linear programming problem is one of the most important branches of operations research, which studies the problem of maximizing (or minimizing) a linear objective under the constraints of a linear inequality or equation. The model is used extensively for planning agricultural cropping structures under multiple conditions [37–39]. In this research, we could use this model to obtain the optimal solution for the acreage of the two crops. The model contains objective functions and constraints. Among them, we mainly constrained the two crops at four levels: Total carbon emission, irrigation water consumption, grain yield, and farm household income. We use Linear Interactive and General Optimizer (LINGO) software to calculate [40]. The constraint model is as follows:

$$F(x)_{min} = C_M X_M + C_W X_W \tag{10}$$

where $F(x)$ represents carbon emissions for both crops; $X_M$ and $X_W$ are the crop area planted with maize and wheat, respectively; and $C_M$ and $C_W$ are the current average carbon emission factor per hectare planted with maize and wheat, respectively.

In order to minimize the irrigation water consumption for agricultural production without affecting the farmers' profitability, we constructed three levels of constraints.

Constraint 1: Total planted area. This study used the areas planted with maize and wheat crops as independent variables. Considering the future urban expansion, the current total planted area was set to the maximum value:

$$X_M + X_W \leq \alpha \tag{11}$$

where $\alpha$ represents the total planted area.

Constraint 2: Irrigation water security. Since the irrigation water consumption per planted area is different for the two crops, the total irrigation water consumption needs to be controlled when performing the irrigation pattern update [41,42]. The expressions are as follows:

$$\beta_M X_M + \beta_W X_W \leq IWC_{Total} \tag{12}$$

where $\beta$ represents the water consumption per unit area of crop cultivation; $IWC_{Total}$ represents total current irrigation water consumption.

Constraint 3: Income security for farmers. When restructuring the regional irrigation system, the stability of farmers' income needs to be guaranteed and the increase in farmers' income is considered as a win–win strategy. Since the policy of Zhangye City encourages the planting of seeded maize and the proportion of seeded maize planting has increased year by year, this study used the income per unit area of maize planting with the price of seeded maize [43]. The expression is as follows:

$$\delta_W X_W + \delta_M X_M = \delta_{Total} \tag{13}$$

where $\delta$ represents farmers' income per unit area of crop cultivation.

Constraint 4: Ensure food security. According to the data of Zhangye City Bureau of Statistics, both studied food crops can be exported to satisfy the region's own self-sufficiency. To ensure regional food security, the production of each grain should not be lower than the region's needs [44]. The expressions are as follows:

$$\varepsilon_W X_W + \varepsilon_M X_M \geq \varepsilon_{Total} \tag{14}$$

where $\varepsilon$ represents the grain yield per unit area; $\varepsilon_{Total}$ represents total food production of both crops.

## 3. Results

### 3.1. Differences in Spatial Patterns of Irrigation Water Consumption

The spatial distribution pattern of irrigation water consumption for maize and wheat in the middle reaches of the Heihe River is depicted in Figure 4. The average irrigation water consumption for maize is 352.56 mm ha$^{-1}$, while that for wheat is 288.85 mm ha$^{-1}$. From the standpoint of water-saving irrigation, it is evident that wheat has a significantly lower average water consumption compared to maize.

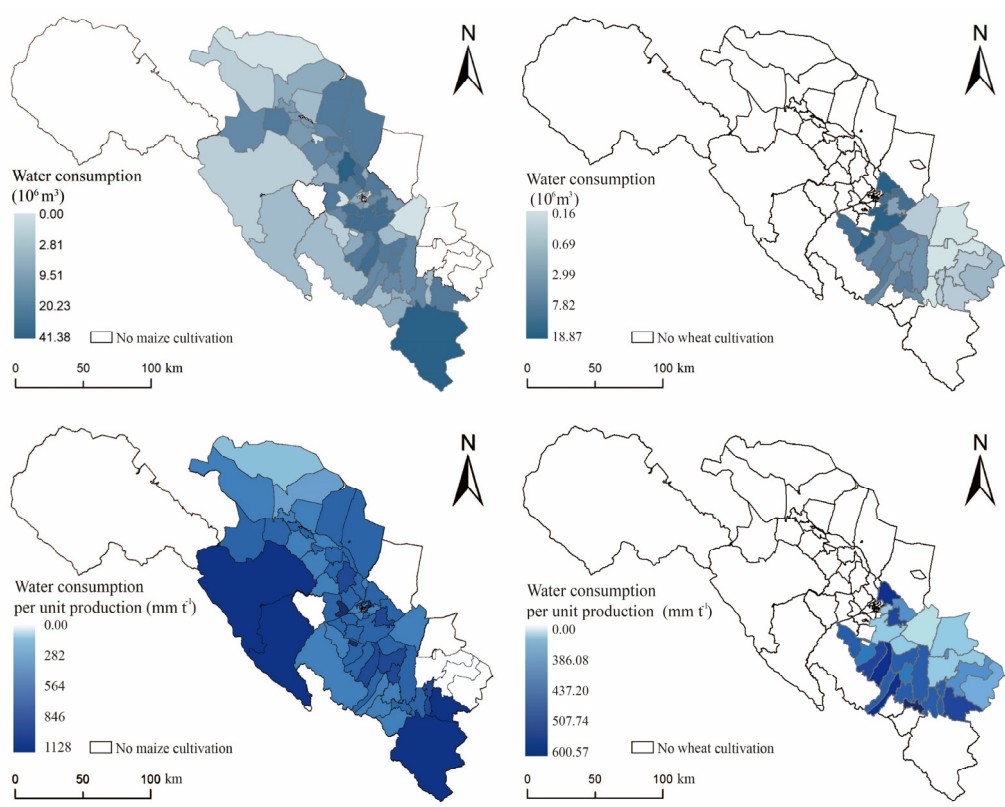

**Figure 4.** Spatial distribution of irrigation water consumption and water consumption per unit of yield for maize and wheat.

In terms of the spatial pattern of water consumption per unit of irrigation, the planting scale for maize covers almost all of middle reaches of the Heihe River basin. However, the levels of irrigation water efficiency vary across the region. The areas with low irrigation water consumption per unit, which indicates high irrigation water use efficiency, are mainly located in the end basins of the tributaries of the Heihe River. These areas also have high topography, with the southwest and southeast of the region being typical examples. On the other hand, the areas with relatively high irrigation water efficiency are mainly found in the planting area near the upstream of the Heihe River and some downstream regions. The upper towns of the Heihe River display the lowest irrigation water consumption across the region.

In the case of wheat, the irrigation water efficiency exhibits a distinctly opposite trend compared to maize. Wheat production in the middle reaches of the Heihe River is concentrated in the upper basin of the river. Spatially speaking, this region can be roughly divided into two areas: The main basin of the Heihe River, which encompasses towns in the southwest of the planting area, and the tributary basins, which include towns in the northeast. According to the results, the irrigation water consumption in the main basin of the Heihe River is significantly higher than that in the tributary basins, with Xintian Township (541 mm t$^{-1}$), Shunhua Township (552 mm t$^{-1}$), and Yonggu Township (600.56 mm t$^{-1}$) reporting particularly high levels of consumption. Interestingly, these towns also exhibit higher efficiency in terms of maize irrigation water use and a higher level of water consumption for maize irrigation. Their irrigation water consumption per unit yield was higher in Xintian Town (728 mm t$^{-1}$), Sunhua Town (551 mm t$^{-1}$), and Yonggu Town (605 mm t$^{-1}$) than the average water consumption of maize (515.6 mm t$^{-1}$).

Regarding variability, maize irrigation water consumption exhibited higher variability than wheat in the middle reaches of the Heihe River. When excluding the four towns with very low maize cultivation in the end basin of the Heihe tributaries, the difference between the highest level of irrigation water consumption in Xintian Township (728 mm t$^{-1}$) and the lowest level in Luocheng Township (169 mm t$^{-1}$) exceeded 550 mm t$^{-1}$. On the other hand, for wheat cultivation, the difference between the highest level of irrigation water consumption in Yonggu Township (600.56 mm t$^{-1}$) and the lowest level (283.76 mm t$^{-1}$) was approximately 320 mm t$^{-1}$. This indicates that maize exhibited higher irrigation water efficiency and greater spatial variation compared to wheat, the latter of which displayed less pronounced differences in irrigation water consumption across the region.

Moreover, for the distribution pattern in the middle reaches of the Heihe River, surface water supply proved to be particularly robust in the upper and middle regions of the basin, highlighting its importance in supporting local agricultural activities.

### 3.2. Differences in Spatial Patterns of Water-Related Carbon Emissions

The spatial distribution of energy consumption for maize and wheat irrigation in the middle reaches of the Heihe River is presented in Figure 5. The average energy consumption for maize irrigation is 42.30 kWh t$^{-1}$, whereas it is 46.84 kWh t$^{-1}$ for wheat. In terms of total regional energy consumption, the high energy consumption for maize irrigation in Zhangye is primarily concentrated in densely populated and intensively planted areas of the main stream basin of the Heihe River. On the other hand, high energy consumption for wheat is mainly concentrated in the towns located in the eastern part of the middle reaches of the Heihe River, followed by some provinces in the planting area near the upstream of the Heihe River. Overall, maize cultivation consumes slightly less energy than wheat.

The spatial characteristics of energy consumption per unit of production for maize irrigation are evident, with the areas of lower regional energy consumption for maize predominantly situated in the planting area near the upstream of the Heihe River, where the average irrigation energy consumption is about 42.26 kwh t$^{-1}$. This is followed by areas surrounding major cities in the middle reaches of the Heihe River, with irrigation energy consumption of about 71.40 kwh t$^{-1}$. The overall distribution of energy consumption displays an inverse relationship with the distance from the main stream of the Heihe River.

As for wheat, its spatial characteristics mirror those of wheat irrigation water efficiency. The average energy consumption in the main stream basin of the Heihe River (46.45 kWh t$^{-1}$) is significantly higher than that in the tributary basins (48.32 kWh t$^{-1}$). However, from the variability perspective, the spatial differences in energy consumption for wheat cultivation are minimal, while spatial heterogeneity is more pronounced for maize.

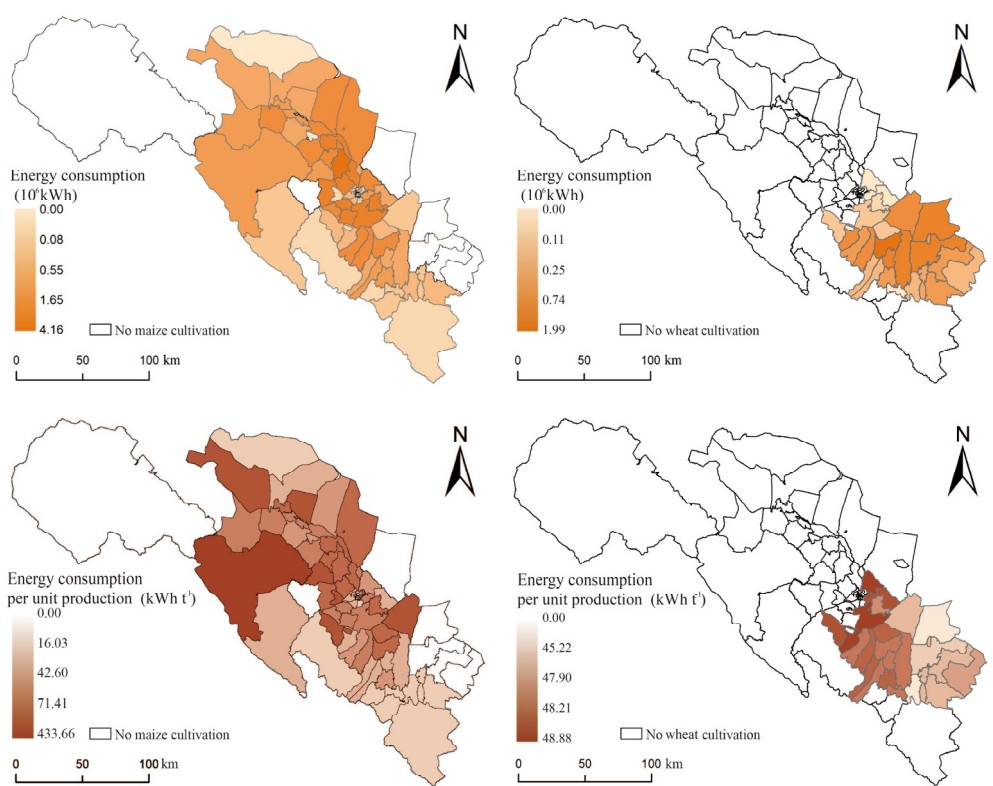

**Figure 5.** Spatial distribution of energy consumption and energy consumption per unit of production for maize and wheat.

### 3.3. Differences in Spatial Patterns of Carbon Emissions from Irrigation

Figure 6 shows the spatial distribution pattern of carbon emissions resulting from maize and wheat irrigation in the middle reaches of the Heihe River. Carbon emissions were obtained using the IPCC coefficient method, which yielded a similar spatial pattern of carbon emissions and energy consumption. Overall, the carbon emission of irrigated maize per unit of production is 18.58 t t$^{-1}$, while that of wheat is 20.88 kWh t$^{-1}$. In general, the carbon emission of maize cultivation is slightly lower than that of wheat.

The carbon emissions of maize are mainly concentrated in the large-scale maize-growing areas surrounding major towns in the middle part of the Heihe River, whereas the lower basin of the Heihe River exhibits higher total carbon emissions than the upper river. Areas with low carbon emissions are mainly concentrated in the middle reaches of the Heihe River region directly south of the river. Regarding the overall spatial pattern of carbon emissions from wheat, the eastern and central towns exhibit higher total carbon emissions, while areas closer to the cities exhibit lower total carbon emissions.

Regarding carbon emissions per unit of production, the differences between the two crops are still evident. For maize, areas with lower carbon emissions per unit of production are mainly concentrated in the upper part of the Heihe River (8.92 t t$^{-1}$). Areas with higher carbon emissions are mainly concentrated in the central part of the region where maize is concentrated and in the lower part of the Heihe River, where carbon emission levels are not very different, with an average carbon emission of about 26.33 t t$^{-1}$ of yield. Lastly, areas with the highest carbon emissions are located in the western and eastern parts of the region in towns not in the main trunk watershed of the Heihe River. The overall spatial pattern of carbon emissions resulting from maize irrigation indicates that the main trunk basin of the Heihe River has the lowest carbon emissions, and the further the distance between maize cultivation and the river, the higher the carbon emissions. Furthermore, the carbon emission level in the planting area near the upstream of the Heihe River is lower than that in the lower reaches.

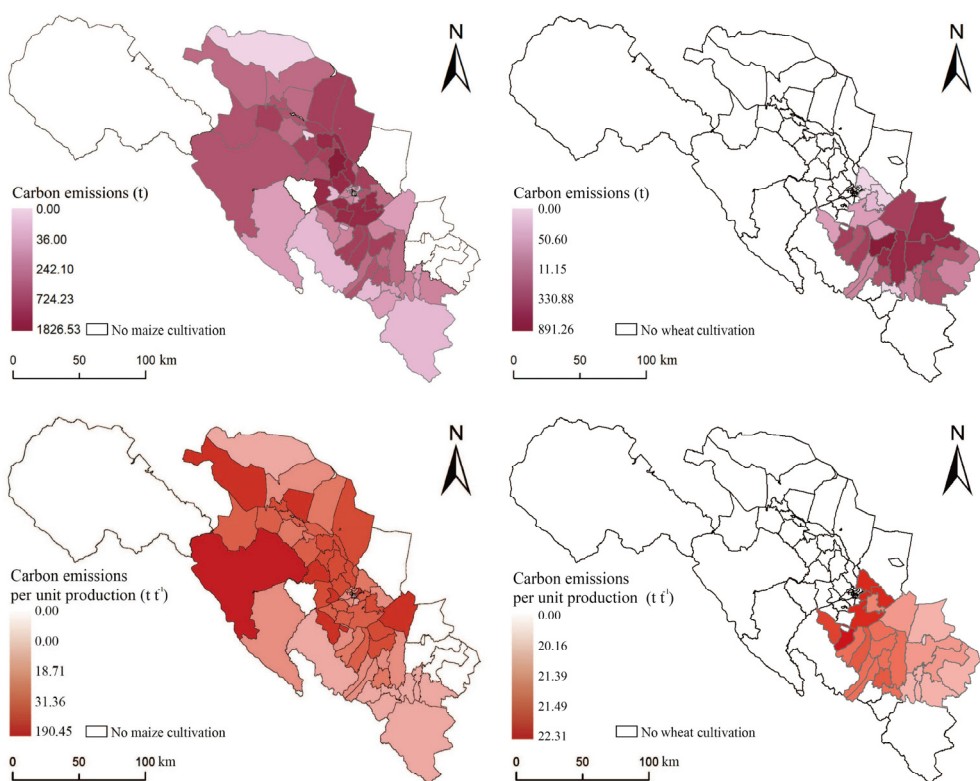

**Figure 6.** Spatial distribution of carbon emissions and carbon emissions per unit of production for maize and wheat.

Regarding the carbon emissions per unit yield of wheat, similar to the spatial pattern of energy consumption, the spatial variability is not significant, and they all lie between 20.16 t t$^{-1}$ and 22.31 t t$^{-1}$. Among them, the average carbon emission level in the main stream basin of the Heihe River is relatively higher than that in the tributary basins. Comparing maize, it can be seen that its carbon emission level is slightly better than that of irrigated maize in the middle and lower reaches of the Heihe River in absolute terms, but higher than that of maize in the planting area near the upstream of the Heihe River. This provides a guideline for future low-carbon agriculture.

### 3.4. Agricultural Planting Structure Adjustment under Linear Programming Model

The current planted areas for maize and wheat are 130,576.54 hectares and 35,627.48 hectares, respectively. Taking into consideration the goals of ensuring food security and securing farmers' income, we conducted spatial optimization model calculations to determine the optimal planting areas for maize and wheat. Our calculations resulted in an optimal maize planting area of 136,203.9 hectares and an optimal wheat planting area of 21,019.93 hectares. Under this model, the area planted with wheat is further optimized to 59% of the current level, while the area planted with maize is expanded to 104% of the current level. This is a side indication of the ecological superiority of maize cultivation over wheat in terms of food security and sustainable agricultural development only. If the cropping structure is updated to the proposed configuration without optimizing the spatial pattern of crops, it would result in a reduction of 1.5% in total irrigation water consumption for both crops and a significant decrease of 31% in total carbon emissions.

## 4. Discussion

### 4.1. A Trade-off among Low-Carbon Agriculture, Irrigation Water Security, and Food Security

Irrigated agriculture is an important means of securing food production in drylands, yet it is also one of the major contributors to regional carbon emissions and water consumption [45–47]. Although the development goal of ensuring food security may seem

irreconcilable with the sustainable agricultural goal of improving resource efficiency in agricultural production processes and reducing carbon emissions, numerous studies have demonstrated that policies can achieve a win–win situation when multiple factors are considered collectively [48–50]. There is often perceived to be a trade-off between ensuring agricultural production and promoting low-carbon agriculture.

The basic requirement for modeling the trade-off between the income of farmers from agricultural cultivation and sustainable agricultural development is to establish a connection between inputs and outputs. Previous studies have shown that the structure of agricultural cultivation is limited by yields, provided that agricultural production is maintained. This has a direct effect on water sacrifice and greenhouse gas emissions [51]. Niu et al.'s research results show that the influence coefficient of cropping type on the agricultural carbon footprint can reach 0.31, which is the highest among various influencing factors [52]. The willingness of farmers to be active is very important for the actual agricultural emission reduction actions. In fact, studies have shown that low-carbon agricultural development has a positive contribution to farm household income [53]. The goals of maintaining farm household income and reducing regional carbon emissions can be achieved simultaneously as long as the agricultural production process is supported by policy and accompanied by precise macro-regulation [54].

Meanwhile, another dimension of food security is to ensure that farmers' incomes do not fall as a result of low-carbon agriculture. Ahumada et al.'s research has demonstrated a strong correlation between the incomes generated by two major food crops: Maize and wheat [7]. In Zhangye City, for example, wheat cultivation decreased by $3.61 \times 10^3$ ha from 2007 to 2012, while maize sowing increased by $39.21 \times 10^3$ ha [55]. During this period, the price of maize also increased more than that of wheat. Updating the cropping structure according to the characteristics of crops can effectively reduce carbon emissions while ensuring the interests of farmers [56]. Based on this premise, when adjusting agricultural acreage for the two crops, we take into full consideration four factors: Irrigation water consumption, carbon emissions, food production, and farm household income. This approach aims to ensure the positive and virtuous development of agricultural production while simultaneously guaranteeing sustainable and optimal agricultural development.

During the past decade, the government of Zhangye City has continuously promoted the optimization of crop cultivation structure, especially for food crops [57,58]. As proven by our study, the decision to expand maize production has effectively achieved the goal of reducing agricultural carbon emissions while safeguarding irrigation water resources. However, on the one hand, our research proves that there is room for further optimization of the cultivation of both crops. The area under maize cultivation can be further expanded. On the other hand, according to our study and previous research results, the location and spatial pattern of planting of both crops need to be adjusted [59].

### 4.2. Spatial Differences in Irrigation Water Consumption of Major Food Crops in Arid Regions

Our research shows that for typical grain crops in the middle reaches of the Heihe River basin, irrigation water consumption is much higher for maize than for wheat. Meanwhile, the two crops have opposite irrigation water consumption patterns in the upper and lower reaches of the Heihe River. In the planting area near the upstream of the Heihe River, surface water resources are abundant, and the irrigation water consumption for maize is low while that for wheat is high. Accordingly, it can be inferred that wheat should be planted more in the lower reaches of the Heihe River and other areas where surface water is not sufficient from the perspective of planting structure because of its low corresponding capacity for surface water. This finding is consistent with the existing research results [60–62]. From the water footprint perspective, the green and blue water footprints of maize were 0.457 and 0.058 $m^3$ $kg^{-1}$, respectively, compared to 0.55 and 0.114 $m^3$ $kg^{-1}$ for wheat [63,64]. This reflects the fact that wheat is more responsive to both precipitation and surface water resources than maize. However, wheat experiences a short precipitation period in its growth cycle and is more dependent on irrigation water [65]. In

Gong et al.'s study, it was shown that 51.88% and 47.36% of the total evapotranspiration of wheat and maize were effectively precipitated [66,67]. In contrast, Wang et al.'s study shows that although increasing food crop cultivation has a positive effect on agricultural irrigation water efficiency, wheat has a larger acreage coefficient and expanding the relative acreage of wheat can save irrigation water consumption [68]. Although for the whole area wheat irrigation energy consumption is slightly higher than that of maize, the energy consumption of maize irrigation is low and that of wheat is high in the middle and near the upstream of the Heihe River where surface water is abundant. For arid areas, both precipitation and surface run-off during the wheat-growing season are low for the year, which leads to groundwater supply as an inevitable choice to achieve the economic effect of wheat in order to balance irrigation water consumption [69]. The results of these studies demonstrate that adjusting the planting area of wheat does not have a significant impact on its yield but can significantly increase the water-saving capacity of regional irrigation systems, provided that the planted area is protected. Based on this, we can conclude that regional wheat is more suitable for planting in areas where surface water and precipitation are scarce, while maize is more suitable for planting in areas where surface water and precipitation are more abundant.

## 5. Conclusions

Within the framework of low-carbon agriculture and water-saving agriculture, an examination of the cropping structure in arid regions holds potential for the optimization of forthcoming agricultural practices. By specifically examining the primary food crops cultivated in the middle reaches of the Heihe River, this study establishes precise county-level water–carbon relationships. It further investigates alterations in the spatial distribution of irrigation water utilization and carbon emissions for these two crops. Additionally, a linear programming model is employed to formulate a low-carbon agricultural cropping structure, taking into account factors such as food security and farmers' income. Our conclusions are as follows: (1) The carbon emissions per unit of water consumption for maize are $0.03 \times 10^{-6}$ t mm$^{-1}$, and $0.49 \times 10^{-6}$ t mm$^{-1}$ for wheat. (2) The spatial distribution of irrigation water allocation for maize and wheat revealed distinctive patterns. Maize cultivation displayed lower demand for irrigation water in the growing areas near the upper reaches of the Heihe River, while higher demand was observed in the lower reaches. Furthermore, carbon emissions from both crops were relatively low in the upstream regions. (3) As a result, the cultivated area for maize expands to 104% its original size, while the area for wheat reduces to 59% its original size. These findings demonstrate that growing maize in areas with relatively abundant surface water for arid regions can help reduce carbon emissions from irrigated agriculture, while wheat cultivation is more suitable for areas with insufficient surface water.

The study also has some limitations, which can be further improved in future studies. Firstly, based on the statistical data from Zhangye City in 2020, it can be observed that the proportion of cereal crops in the middle reaches of the Heihe River is as follows: Maize accounts for 56%, wheat accounts for 17%, and potatoes account for 12%. However, due to the scattered distribution of potato cultivation areas, it is not possible to present their spatial distribution levels. In the future, we can enhance technological advancements to enable a more comprehensive assessment of all cereal crops. Secondly, the primary focus of this study was to establish a water–carbon nexus in the middle reaches of the Heihe River. While the agricultural cultivation data used in this study pertain to the year 2020, it should be acknowledged that the structure and scale of agricultural cultivation in this region can vary significantly from year to year. Therefore, future studies should aim to accumulate year-by-year data to enable longitudinal investigations and derive more consistent conclusions. Finally, this study proposed the optimization of the cropping structure for the two crops using a linear programming model. Nevertheless, the response of these crops to diverse irrigation water consumption and carbon emission environments in arid areas can differ. Therefore, it is recommended that future research incorporates

optimization strategies for crop planting structures within geographic space to account for these variations.

Based on these findings, we make the following recommendations for developing low-carbon agriculture in arid regions: (1) In Zhangye City, the introduction of the "Implementation Plan for the Circular Development of Maize in Zhangye City" has promoted the scaled cultivation of hybrid maize. As a result, the area under maize cultivation in Zhangye has been increasing in recent years, while the area under other cereal crops has been further reduced. Our study suggests that the current policy is moving toward promoting agricultural production. In the future, it is recommended to build on this policy to further encourage maize cultivation. (2) Considering China's policy of maintaining arable land without reduction, in the middle reaches of the Heihe River, the restructuring of maize and wheat cultivation is not simply a matter of interchanging between these two crops. By slightly increasing maize cultivation, the scale of wheat cultivation can be significantly reduced to ensure food security and farmers' income. The fertile arable land can be utilized for growing cash crops, thereby enhancing farmers' income. This study is valuable for the future ecologically sustainable development of food crop cultivation in arid regions.

**Author Contributions:** Methodology, B.L.; Formal analysis, B.L.; Investigation, B.L.; Writing—original draft, B.L.; Writing—review & editing, M.N., J.Z., X.Z., R.C., X.L., J.L. and Y.W.; project administration, M.N.; Funding acquisition, M.N. All authors have read and agreed to the published version of the manuscript.

**Funding:** This research was funded by the National Natural Science Foundation of China under Grant 52208041, the Key Laboratory of Ecology and Energy-saving Study of Dense Habitat (Tongji University), Ministry of Education under Grant 20220110 and the Beijing High-Precision Discipline Project, Discipline of Ecological Environment of Urban and Rural Human Settlements, the National Key Research and Development of China, grant number 2022YFC380260403 and the China Urban Construction Design and Research Institute, grant number Y06Y22006.

**Data Availability Statement:** The data are not publicly available due to the team will conduct further research on the area and the data are kept confidential for the time being.

**Conflicts of Interest:** The authors declare no conflict of interest.

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
