# Peer review of "Agricultural Cultivation Structure in Arid Areas Based on Water–Carbon Nexus—Taking the Middle Reaches of the Heihe River as an Example"

_land, doi:10.3390/land12071442_

Round 1
Reviewer 1 Report
Thank you for inviting me to review the manuscript on "Research on Agricultural Cultivation Structure in Arid Areas Based on Water-Carbon nexus-taking the middle reaches of the Heihe River as an example". The author studied the irrigation water use efficiency and carbon emission differences of crops in arid areas and their spatial distribution using wheat and maize. I think this article is an interesting and meaningful, but there are some issues that need to be addressed.
1.I suggest the author can supplement some specific aims in the last paragraph of introduction section.
2.In this paper, the author has utilized some different types data; however, it remains unclear how these data sets correspond with one another in terms of scale.
3.In Figure1, please give us more description. By the way, I think that is land use and land cover classification map, which year of this classification map?
4.In Methods and data section, I suggest the author first to introduce data and then introduce the methods. In addition, the author should be supplementing period and some details about these images. How much accuracy of verified?
5.Please added some references about the equations. Where show the Figure 2 text in the paper?
6.Please note the Figure 5 scale unit, km or kilometers
7.Please tell us this paper limitations and outlook in Discussion section.
8.Please concise the conclusion section. Some conclusions should be deleted, Because the conclusion and the summary share some of the same expressions.
However, I believe this manuscript has a potential publish value. I suggest the author can be major revised it thoroughly.
Author Response
Reviewer #1
Reviewer #1: Thank you for inviting me to review the manuscript on "Research on Agricultural Cultivation Structure in Arid Areas Based on Water-Carbon nexus-taking the middle reaches of the Heihe River as an example". The author studied the irrigation water use efficiency and carbon emission differences of crops in arid areas and their spatial distribution using wheat and maize. I think this article is an interesting and meaningful, but there are some issues that need to be addressed.
Response:
Thank you very much for your recognition of the article! Your opinions helped to improve academic rigor of our article.
The italic and underlined paragraphs are comments made by reviewers
The paragraphs with green colors are unchanged paragraphs in the original text.
The paragraphs with red colors are revised or added paragraphs in the original text.
Point 1: I suggest the author can supplement some specific aims in the last paragraph of introduction section.
Response 1: Thank you for your suggestion, according to your suggestion, we added the information you suggested. We have added to the last paragraph of the introduction section. The modified paragraphs are as follows:
Therefore, this research aims to establish the spatial distribution data of major food crops, namely maize and wheat, in the middle reaches of the Heihe River with town-level accuracy. This is accomplished by employing the Water-Carbon Nexus framework in a typical arid zone. In this paper, town-level data on irrigation water consumption and agricultural carbon emissions are obtained through farmer surveys, meteorological data, and satellite data. These data are then utilized to calculate the carbon emission intensity per unit of water consumption, thereby enabling the evaluation of the carbon emission capacity of crops. Subsequently, a comprehensive evaluation was conducted, which integrated considerations of food security, farm household income, and agroecological values. In the current global context, there is a pressing need for concerted efforts to mitigate the adverse impact of agriculture on the climate and ensure food production. Under the guidance of the United Nations Sustainable Development Goals (SDGs), the pursuit of a secure water environment, low-carbon emission environmental protection, sustainable agricultural development, and food security are globally unified objectives. The findings offer valuable guidance for the cultivation of food crops in arid regions worldwide, The research findings can serve as guidance for the cultivation of food crops in arid areas facilitate the exploration of spatial distribution patterns of water consumption and carbon emissions among different crops, and offer valuable insights towards achieving an optimal crop cultivation pattern that ensures food security.
Point 2: In this paper, the author has utilized some different types data; however, it remains unclear how these data sets correspond with one another in terms of scale.
Response 2: Thank you for your suggestion. In order to better construct the scale correlation between multiple elements, we have added the paragraph "2.2.2 Data integration at different scales", and added a schematic diagram. The details are as follows:
2.2.2. Data integration at different scales
In this study, we used different sources of data with different precision. For ex-ample, satellite data are from open-source websites and their data type is raster data; meteorological data are attribute data with precision to different counts; survey data are point data. In order to obtain geographic data with precision to county, we use various spatial processing techniques, such as Euclidean distance, spatial algorithms, etc. The processing of different data is shown in Figure 3.
Figure 3. Research framework diagram
Point 3: In Figure1, please give us more description. By the way, I think that is land use and land cover classification map, which year of this classification map?
Response 3: We totally agree with your suggestion. The clarity of our statement regarding figure 1 appears to be inadequate. In response, we have incorporated more precise latitude, longitude, and figure name information into the image. Below, we present the figure 1, both before and after the modifications are as follows ( (please see the attachment)):
(Before)Figure 1. Zhangye City Location and Land Use Data
(After)Figure 1. Zhangye City Location and Land Use Data
Point 4: In Methods and data section, I suggest the author first to introduce data and then introduce the methods. In addition, the author should be supplementing period and some details about these images. How much accuracy of verified?
Response 4: Thank you for your suggestion. We move the presentation of the data from paragraph 2.2.6. to paragraph 2.2.1. We have added accuracy notes to the images. For details, see the changed Figure 1. In paragraph 2.2 Methods and Data, we are not precise in our presentation of some of the data. In addition, in some of the method statements, we have added detailed descriptions of the data and added descriptions of the accuracy. For example:
To begin with, Landsat 8 data was employed to obtain the Normalized Difference Vegetation Index (NDVI) for the region. Through radiometric correction, atmospheric correction, and image cropping using the ENVI software, the precise NDVI temporal curve of the region was derived. ENVI software is developed by Exelis Visual Information Solutions, the main purpose of this software is to process and analyze remote sensing data. The NDVI values were calculated for each year and time period, resulting in the generation of NDVI change curves for each year.
To ensure the accuracy of the planting area distribution data obtained from satellite imagery in this study, it is necessary to conduct a thorough verification. Several evaluation methods were employed to validate the NDVI (Normalized Difference Vegetation Index) data, including kappa coefficients, confusion matrix analysis, squared mean error, and normalized squared mean error. These techniques were chosen to assess the reliability and precision of the obtained data.After years of verification using NDVI data, the overall classification accuracy of Landsat 8 data for the planting area of the two crops we measured, employing segmented multispectral analysis, has been found to approach approximately 89%.
To determine the irrigation water consumption (IWC) for weight loss, the actual irrigation water consumption of two crops was measured using Meteorological Bureau data and Meteorological detection data. A comprehensive calculation process is employed, involving three main steps. The first step involves calculating evapotranspiration (ETc) and effective precipitation (EP) using meteorological data and information on the area dedicated to maize planting.
Point 5: Please added some references about the equations. Where show the Figure 2 text in the paper?
Response 5: Thank you for your suggestion. We have added "The detailed flow chart is shown in Fig. 2" in the 2nd and 2nd paragraphs. In addition, we have added ample references to the Penman-Monteith formula, the use of USDA SCS, and the use of the linear programming in agricultural farming structures to support the use of these formulas. The use of linear programming is described in detail. The introduction is as follows:
The linear programming problem is one of the most important branches of operations research, which studies the problem of maximizing (or minimizing) a linear objective under the constraints of a linear inequality or equation. The model is used extensively for planning agricultural cropping structures under multiple conditions [29-31]. In this research we can use this model to obtain the optimal solution for the acreage of two crops.
- ACS S, BERENTSEN P B M, HUIRNE R B M. Conversion to organic arable farming in The Netherlands: A dynamic linear programming analysis [J]. Agricultural Systems, 2007, 94(2): 405-15, doi:https://doi.org/10.1016/j.agsy.2006.11.002.
- BAUER S, KASNAKOGLU H. Non-linear programming models for sector and policy analysis: Experiences with the Turkish agricultural sector model [J]. Economic Modelling, 1990, 7(3): 275-90, doi: https://doi.org/10.1016/0264-9993(90)90013-T.
- BARTOLINI F, BAZZANI G M, GALLERANI V, et al. The impact of water and agriculture policy scenarios on irrigated farming systems in Italy: An analysis based on farm level multi-attribute linear programming models [J]. Agricultural Systems, 2007, 93(1): 90-114, doi: https://doi.org/10.1016/j.agsy.2006.04.006.
Point 6: Please note the Figure 5 scale unit, km or kilometers
Response 6: Thank you for your offer. It was an oversight on our part. The altered figure are as follows (please see the attachment):
Point 7: Please tell us this paper limitations and outlook in Discussion section.
Response 7: Thank you for your suggestion. We added this paragraph to the concluding paragraph of the paper to express our limitations of the paper and our vision for the future:
The findings presented in this paper are undoubtedly valuable, yet they also pos-sess certain limitations that could be further refined in future research. Firstly, based on the statistical data from Zhangye City in 2020, it can be observed that the proportion of cereal crops in the middle reaches of the Heihe River is as follows: maize accounts for 56%, wheat accounts for 17%, and potatoes account for 12%. However, due to the scattered distribution of potato cultivation areas, it is not possible to present their spa-tial distribution levels. In the future, we can enhance technological advancements to enable a more comprehensive assessment of all cereal crops. Secondly, the primary focus of this study was to establish a water-carbon nexus in the middle reaches of the Heihe River. While the agricultural cultivation data used in this study pertains to the year 2020, it should be acknowledged that the structure and scale of agricultural culti-vation in this region can vary significantly from year to year. Therefore, future studies should aim to accumulate year-by-year data to enable longitudinal investigations and derive more consistent conclusions. Finally, this study proposed the optimization of the cropping structure for the two crops using a linear programming model. Nevertheless, the response of these crops to diverse irrigation water consumption and carbon emission environments in arid areas can differ. Therefore, it is recommended that future research incorporates optimization strategies for crop planting structures within geographic space to account for these variations.
Point 8: Please concise the conclusion section. Some conclusions should be deleted, Because the conclusion and the summary share some of the same expressions.
Response 8: Thank you for your suggestion. We made substantial changes to the concluding paragraph to reduce repetition with the previous text and to refine the language. The following is from the first paragraph of our concluding paragraph:
Within the framework of low-carbon agriculture and water-saving agriculture, an examination of the cropping structure in arid regions holds potential for the optimization of forthcoming agricultural practices. This study directs its attention to the principal food crops, namely maize and wheat, cultivated in the middle reaches of the Heihe River. The research involves the construction of a county-precision water-carbon nexus, exploring the spatial pattern variability of irrigation water consumption and carbon emissions for these two crops. Additionally, an agricultural cropping structure is established using a linear programming model to ensure the most economical allocation of irrigation water and the lowest carbon emissions, while simultaneously addressing food security and farmers' income considerations. Our conclusions are as follows: (1) The carbon emissions per unit of water consumption for maize are 0.03×10-6t mm-1, and 0.49×10-6t mm-1for wheat. The carbon emissions per unit of irrigation water consumed by maize cultivation are lower than those of wheat. (2) The spatial distribution patterns of irrigation water consumption were opposite for maize and wheat. The former has lower irrigation water consumption in the planting area near upstream. of the Heihe River and higher in the lower reaches. In contrast, the water consumption pat-tern of wheat irrigation is the opposite. The main grain crops in the middle reaches of the Heihe River basin are maize and wheat. From the perspective of water saving, spring wheat cultivation exhibits more water saving ability in the arid zone. However, the carbon emission is 18.58t t-1, while for wheat, the carbon emission is 20.88t t-1. On average, The carbon emission of maize is lower than that of wheat cultivation. In terms of spatial patterns. In areas with relatively abundant surface water in the planting area near upstream of the Heihe River, the irrigation water consumption per unit of production is lower for maize, while the irrigation water consumption is higher for wheat. (3) After taking into account food security and energy conservation, the optimal cropping structure is 136203.9 ha of maize and 21019.93 ha of wheat. As a result, the cultivated area for maize expands to 104% of its original size, while the area for wheat reduces to 59% of its original size. The area planted with maize can be appropriately increased, while the area planted with wheat can be appropriately reduced under the premise of food security.

Reviewer 2 Report
The topic of dryland cultivation is very important in view of the realities that await mankind in the future. I.e. the more significant impacts of climate change on food security. There is a need to respond to rising temperatures and decreasing rainfall - or its shift to other parts of the world.
Minor formal and substantive comments:
Lines 12-25: Abstract: Result (1): If the authors provide specific results of carbon emissions in the abstract, it is appropriate to also provide specific results of water consumption for irrigation . Result (3): In the abstract, the authors report the optimal crop mix (maize and wheat) in hectares. It would be useful to indicate how the authors' solution differs from the real situation in the region (also in percentage terms, not just in value terms).
Lines 79-93: The introduction (last paragraph) should not contain methodological procedures. This is the purpose of the next chapter.
Line 111: What complex crops does the author [22] have in mind?
Line 119: Which national authorities?
Line 180: Add a reference to the source of the Penman-Monteith equation (UN).
Line 190: appropriate to add reference to the USDA SCS model (in the methodology and literature sources).
Line 297: Results: the title of the paper states that the authors will conduct the study on the middle Heike River. The results report information for the upper reaches. This does not contradict the research, but it is sometimes confusing.
Line 294: Section 2.2.6 should indicate the area farmed by the farmers contacted. And what percentage of the area is on the middle Heike River.
Line 407: There is no reference to the formula (procedure) according to which the authors optimized the seed area. How will the optimization have a specific impact on water management and how does carbon emissions change (suggestion versus reality).
The conclusion should bring out the best. What is new and beneficial. It is not appropriate to repeat what is stated in the methodology. Proposals for changes in cropping areas should also be given as a percentage (proposal versus reality).
Recommendations (1) and (2) are general. These recommendations can be made without extensive research. For example, it is also important to conclude by stating the extent to which the proposals are feasible in relation to food security and farmers' incomes.
-
Author Response
Reviewer #2
Reviewer #2: The topic of dryland cultivation is very important in view of the realities that await mankind in the future. I.e. the more significant impacts of climate change on food security. There is a need to respond to rising temperatures and decreasing rainfall - or its shift to other parts of the world.
Response:
Thank you very much for your recognition of the article's research purpose! Your comments help to improve the academic rigor of our articles.
The italic and underlined paragraphs are comments made by reviewers
The paragraphs with green colors are unchanged paragraphs in the original text.
The paragraphs with red colors are revised or added paragraphs in the original text.
Point 1: Lines 12-25: Abstract: Result (1): If the authors provide specific results of carbon emissions in the abstract, it is appropriate to also provide specific results of water consumption for irrigation . Result (3): In the abstract, the authors report the optimal crop mix (maize and wheat) in hectares. It would be useful to indicate how the authors' solution differs from the real situation in the region (also in percentage terms, not just in value terms).
Response 1: Thank you for your suggestion. Your suggestions for the summary section of our article are to the point. In response to your comments, we have added a description of irrigation water consumption and replaced the crop structure adjustment with a percentage display. The specific changes are as follows:
Abstract: China faces challenges of food security and sustainable agricultural production. However, current researches rarely address the spatial distribution patterns of water consumption and carbon emissions and its spatial distribution. We studied the irrigation water use efficiency and carbon emission differences of crops in arid areas and their spatial distribution using wheat and maize, two major food crops in the middle reaches of the Heihe River, as examples. Furthermore, we have optimized low carbon cropping of crops under the multiple objectives of water conservation and economic development. The results show that: (1) The carbon emissions per unit of water consumption for maize are 0.03×10-6t mm-1, and 0.49×10-6t mm-1 for wheat. Irrigation water consumption per unit yield is 515.6 mm t-1 for maize and 426.7 mm t-1 for wheat (2) The spatial distribution patterns of irrigation water consumption were opposite for maize and wheat. The former has lower irrigation water consumption in the upper reach’s upstream region of a river’s middle section. of the Heihe River and higher in the lower reaches. In contrast, the water consumption pattern of wheat irrigation is the opposite. (3) After optimizing the cropping mix for both crops, the area planted to wheat should be reduced to 59% of the current size, while maize should be expanded to 104%. The results of the research can effectively guide the future grain crop planting pattern for water-saving agriculture and low-carbon agriculture development in global the arid zone.
Point 2: Lines 79-93: The introduction (last paragraph) should not contain methodological procedures. This is the purpose of the next chapter.
Response 2: Thank you for your suggestion. We did add too many statements about the method. Our changes to the last paragraph of the introduction are as follows:
Therefore, this research aims to establish the spatial distribution data of major food crops, namely maize and wheat, in the middle reaches of the Heihe River with town-level accuracy. This is accomplished by employing the Water-Carbon Nexus framework in a typical arid zone. In this paper, town-level data on irrigation water consumption and agricultural carbon emissions are obtained through farmer surveys, meteorological data, and satellite data. These data are then utilized to calculate the carbon emission intensity per unit of water consumption, thereby enabling the evaluation of the carbon emission capacity of crops. Subsequently, a comprehensive evaluation was conducted, which integrated considerations of food security, farm household income, and agroecological values. In the current global context, there is a pressing need for concerted efforts to mitigate the adverse impact of agriculture on the climate and ensure food production. Under the guidance of the United Nations Sustainable Development Goals (SDGs), the pursuit of a secure water environment, low-carbon emission environmental protection, sustainable agricultural development, and food security are globally unified objectives. The findings offer valuable guidance for the cultivation of food crops in arid regions worldwide, The research findings can serve as guidance for the cultivation of food crops in arid areas facilitate the exploration of spatial distribution patterns of water consumption and carbon emissions among different crops, and offer valuable insights towards achieving an optimal crop cultivation pattern that ensures food security.
Point 3: Line 111: What complex crops does the author [22] have in mind?
Response 3: We reconsidered the reference to [22]. We have corrected the presentation:
However, over the past decade, the middle reaches of the Heihe River have witnessed an expansion of oases aimed at enhancing the scope of agricultural cultivation. Approximately 100,000 hectares of irrigated farmland have been reclaimed, leading to a fragmented arable landscape within the intricate topography of the Middle Heihe River region. Consequently, this has resulted in a lack of scalability in crop cultivation However, the region’s challenging topography and fragmented arable land have resulted in intricate cropping patterns in Zhangye [22].
- Lu, L.; Li, X.; Cheng, G. Landscape evolution in the middle Heihe River Basin of north-west China during the last decade. Journal of Arid Environments 2003, 53, 395-408, doi:https://doi.org/10.1006/jare.2002.1032.
Point 4: Which national authorities?
Response 4: Thank you for raising this issue. We have added to the corresponding statement:
In this study, the planting distribution and yield distribution of two crops in the middle reaches of the Heihe River were determined using Landsat and MOD13 satellite data, along with Digital Terrain Model (DEM) elevation data and land use data provided by the China Geospatial Data Cloud. Finally, the accuracy of the results was validated using agricultural data from China and Zhangye City Statistical Bureau. The accuracy of the results was verified using data from national and local statistical bureaus.
Point 5: Line 180: Add a reference to the source of the Penman-Monteith equation (UN).
Response 5: Thank you for raising this issue. We have added references to it:
In this study, the Penman-Monteith equation is utilized to calculate ETp. This equation approximates net evaporation using meteorological data as a substitute for direct measurement. Widely adopted, the equation is derived by the Food and Agri-culture Organization of the United Nations for simulating potential evapotranspiration [26,27].
- NYOLEI D, DIELS J, MBILINYI B, et al. Evapotranspiration simulation from a sparsely vegetated agricultural field in a semi-arid agro-ecosystem using Penman-Monteith models [J]. Agricultural and Forest Meteorology, 2021, 303: 108370, doi: https://doi.org/10.1016/j.agrformet.2021.108370.
- GAVILáN P, BERENGENA J, ALLEN R G. Measuring versus estimating net radiation and soil heat flux: Impact on Penman–Monteith reference ET estimates in semiarid regions [J]. Agricultural Water Management, 2007, 89(3): 275-86, doi: https://doi.org/10.1016/j.agwat.2007.01.014.
Point 6: Line 190: appropriate to add reference to the USDA SCS model (in the methodology and literature sources).
Response 6: Thank you for raising this issue. We have added references to it:
Empirical models are commonly used to calculate EP based on meteorological precipitation data. In this study, the calculation model utilized is the United States Department of Agriculture Soil Conservation Service (USDA SCS) model [28].
- ZAREI A R, MOGHIMI M M. Modified version for SPEI to evaluate and modeling the agricultural drought severity [J]. International journal of biometeorology, 2019, 63: 911-25, doi: https://doi.org/10.1007/s00484-019-01704-2.
Point 7: Line 297: Results: the title of the paper states that the authors will conduct the study on the middle Heike River. The results report information for the upper reaches. This does not contradict the research, but it is sometimes confusing.
Response 7: Thank you for bringing up the poorly presented parts of our study. What we actually wanted to express was the upstream part of the middle reaches of the Heihe River. Therefore, we have changed all the expressions in the text about the upper reaches from “upper reaches” to “the planting area near upstream”.
Point 8: Line 294: Section 2.2.6 should indicate the area farmed by the farmers contacted. And what percentage of the area is on the middle Heike River.
Response 8: Thank you for your suggestion. We do need further description of the agricultural survey data. The changes we made are as follows:
Survey data is mainly derived from individual interviews conducted with local farmers. In order to demonstrate the significant influence of topographic factors and the distribution of irrigation water resources on the irrigation process, this study specifically selected four representative villages in the middle reaches of the Heihe River Basin, where maize and wheat cultivation is predominant. Each farming household cultivates a non-separable area of 1-2 hectares for both crops. The proportional distribution of major cereal crop cultivation areas in each county is presented in Table 1. These villages are situated in di-verse geographical and topographical settings. The distribution of questionnaires took place in February 2023, resulting in a collection of 421 completed questionnaires. Among them, 409 questionnaires were deemed qualified, while 12 were considered unqualified, yielding a qualification rate of 97.1%. To account for the significant influence of terrain factors and the distribution of irrigation water resources on the irrigation process, this study specifically selects four representative villages situated in distinct geographical and terrain features within the primary food crop growing area of the middle reaches of the Heihe River basin. The survey questionnaire was distributed in February 2023, resulting in the collection of a total of 421 questionnaires. Among these, 409 questionnaires were deemed qualified, while 12 were considered unqualified, yielding a qualification rate of 97.1%.
Table 1. The proportional distribution of major cereal crop cultivation areas.
|
Point Number |
Wheat |
Corn |
Tubers |
|
Zhangye City (Total) |
0.17 |
0.48 |
0.10 |
|
Ganzhou District |
0.04 |
0.92 |
0.00 |
|
Sunan County |
0.14 |
0.27 |
0.03 |
|
Minle County |
0.38 |
0.13 |
0.31 |
|
Linze County |
0.01 |
0.97 |
0.00 |
|
Gaotai County |
0.10 |
0.65 |
0.04 |
|
Shandan County |
0.42 |
0.03 |
0.23 |
Point 9: Line 407: There is no reference to the formula (procedure) according to which the authors optimized the seed area. How will the optimization have a specific impact on water management and how does carbon emissions change (suggestion versus reality).
Response 9 Thank you for your suggestion. We have added formulas for Chapter 2.This paragraph really needs to be supplemented with the results on the actual impact of irrigation water consumption and carbon emissions after planting structure optimization. Our additions are as follows:
Constraint 4: Ensure food security. According to the data of Zhangye City Bureau of Statistics, both studied food crops can be exported to satisfy the region's own self-sufficiency. To ensure regional food security, the production of each grain should not be lower than the region's needs [36]. The expressions are as follows:
(14)
Where represents the Grain yield per unit area; represents total food production of both crops.
The current planted areas for maize and wheat are 130576.54 hectares and 35627.48 hectares, respectively. Taking into consideration the goals of ensuring food security and securing farmers' income, we conducted spatial optimization model calculations to determine the optimal planting areas for maize and wheat. Our calculations resulted in an optimal maize planting area of 136203.9 hectares and an optimal wheat planting area of 21019.93 hectares. Under this model, the area planted to wheat is further optimized to 59% of the current level, while the area planted to corn maize is expanded to 104% of the current level. This is a side indication of the ecological superiority of maize cultivation over wheat in terms of food security and sustainable agricultural development only. If the cropping structure is updated to the proposed con-figuration without optimizing the spatial pattern of crops, it would result in a reduction of 1.5% in total irrigation water consumption for both crops and a significant de-crease of 31% in total carbon emissions.
Point 10: The conclusion should bring out the best. What is new and beneficial. It is not appropriate to repeat what is stated in the methodology. Proposals for changes in cropping areas should also be given as a percentage (proposal versus reality).
Recommendations (1) and (2) are general. These recommendations can be made without extensive research. For example, it is also important to conclude by stating the extent to which the proposals are feasible in relation to food security and farmers' incomes.
Response 10: Thank you for your suggestion. The conclusion section of our previous discussion lacks conciseness and precision. In the revised concluding section, we have made several improvements. Firstly, we have streamlined our statement of methodology to ensure its clarity and accuracy. Secondly, we have enhanced the expression of the study's purpose and significance, providing a more succinct overview of our objectives and the broader implications of our findings. Furthermore, we have updated our policy recommendations to reflect the latest developments and insights derived from our study. Additionally, we have included an assessment of deficiencies and proposed future research directions. The updated version of our conclusion is as follows:
Within the framework of low-carbon agriculture and water-saving agriculture, an examination of the cropping structure in arid regions holds potential for the optimization of forthcoming agricultural practices. This study directs its attention to the principal food crops, namely maize and wheat, cultivated in the middle reaches of the Heihe River. The research involves the construction of a county-precision water-carbon nexus, exploring the spatial pattern variability of irrigation water consumption and carbon emissions for these two crops. Additionally, an agricultural cropping structure is established using a linear programming model to ensure the most economical allocation of irrigation water and the lowest carbon emissions, while simultaneously addressing food security and farmers' income considerations. Our conclusions are as follows: (1) The carbon emissions per unit of water consumption for maize are 0.03×10-6t mm-1, and 0.49×10-6t mm-1for wheat. The carbon emissions per unit of irrigation water consumed by maize cultivation are lower than those of wheat. (2) The spatial distribution patterns of irrigation water consumption were opposite for maize and wheat. The former has lower irrigation water consumption in the planting area near upstream. of the Heihe River and higher in the lower reaches. In contrast, the water consumption pat-tern of wheat irrigation is the opposite. The main grain crops in the middle reaches of the Heihe River basin are maize and wheat. From the perspective of water saving, spring wheat cultivation exhibits more water saving ability in the arid zone. However, the carbon emission is 18.58t t-1, while for wheat, the carbon emission is 20.88t t-1. On average, The carbon emission of maize is lower than that of wheat cultivation. In terms of spatial patterns. In areas with relatively abundant surface water in the planting area near upstream of the Heihe River, the irrigation water consumption per unit of production is lower for maize, while the irrigation water consumption is higher for wheat. (3) After taking into account food security and energy conservation, the optimal cropping structure is 136203.9 ha of maize and 21019.93 ha of wheat. As a result, the cultivated area for maize expands to 104% of its original size, while the area for wheat reduces to 59% of its original size. The area planted with maize can be appropriately increased, while the area planted with wheat can be appropriately reduced under the premise of food security.
The findings presented in this paper are undoubtedly valuable, yet they also possess certain limitations that could be further refined in future research. Firstly, based on the statistical data from Zhangye City in 2020, it can be observed that the proportion of cereal crops in the middle reaches of the Heihe River is as follows: maize accounts for 56%, wheat accounts for 17%, and potatoes account for 12%. However, due to the scattered distribution of potato cultivation areas, it is not possible to present their spatial distribution levels. In the future, we can enhance technological advancements to enable a more comprehensive assessment of all cereal crops. Secondly, the primary focus of this study was to establish a water-carbon nexus in the middle reaches of the Heihe River. While the agricultural cultivation data used in this study pertains to the year 2020, it should be acknowledged that the structure and scale of agricultural cultivation in this region can vary significantly from year to year. Therefore, future studies should aim to accumulate year-by-year data to enable longitudinal investigations and derive more consistent conclusions. Finally, this study proposed the optimization of the cropping structure for the two crops using a linear programming model. Nevertheless, the response of these crops to diverse irrigation water consumption and car-bon emission environments in arid areas can differ. Therefore, it is recommended that future research incorporates optimization strategies for crop planting structures within geographic space to account for these variations.
Based on these findings, we make the following recommendations for develop-ing low-carbon agriculture in arid regions: (1) In Zhangye City, the introduction of the "Implementation Plan for the Circular Development of Maize in Zhangye City" has promoted the scaled cultivation of hybrid maize. As a result, the maize cultivation area in Zhangye City has been increasing in recent years, while the cultivation area of other cereal crops has decreased further. Our research indicates that, for arid regions like the middle reaches of the Heihe River, the average water consumption and carbon emissions associated with wheat cultivation are higher compared to maize cultivation. This suggests that the current policy is moving in the right direction towards promoting agricultural production while ensuring food security. In the future, it is recommended to further encourage maize cultivation based on this policy. (2) In arid regions, to ensure food security and farmers' income, increasing the proportion of maize cultivation compared to wheat cultivation is more effective in reducing irrigation water consumption and carbon emissions. (3) Considering China's policy of maintaining arable land without reduction, in the middle reaches of the Heihe River, the restructuring of maize and wheat cultivation is not simply a matter of interchanging between these two crops. By slightly increasing maize cultivation, the scale of wheat cultivation can be significantly reduced to ensure food security and farmers' income. The fertile arable land can be utilized for growing cash crops, thereby enhancing farmers' income. (1) To improve irrigation water use efficiency, wheat should be grown in areas with relatively insufficient surface water, while maize can be grown in areas with abundant surface water. (2) If the actual food demand is ignored, increasing the proportion of maize cultivation in areas with abundant surface water can help reduce carbon emissions, while wheat cultivation is more desirable for areas with insufficient surface water. This study is valuable for the future ecologically sustainable development of food crop cultivation in arid regions.

Reviewer 3 Report
Dear Editor
After detailed readings in the manuscript, entitled: "Research on Agricultural Cultivation Structure in Arid Areas Based on Water-Carbon nexus _ taking the middle reaches of the Heihe River as an example", the need for studies on food security and sustainable agricultural production is understood. The results of this research can effectively guide the future grain planting pattern towards water-saving and low-carbon agriculture in the arid zone. I suggest ACCEPT the manuscript with minor corrections:
1 - I suggest modifying the title: "Research on Agricultural Cultivation Structure in Arid Areas Based on Water-Carbon nexus _ taking the middle reaches of the Heihe River as an example", for "Agricultural Cultivation Structure in Arid Areas Based on Water-Carbon nexus _ taking the middle reaches of the Heihe River as an example".
2 - At the end of the Abstract, it is necessary to address the importance and need for this study on a global scale.
3 - The introduction is very well reasoned, but I suggest adding the importance of this study to the world at the end of the introduction. This would arouse the interest of readers more.
4 - In Figure 1 can insert the coordinate grid. allowing other researchers to have the exact location of the study area.
5 - The methodology is well founded. The authors really did a good job. Congratulations.
6 - I suggest deleting the subtitle: "3.4. Differences in spatial patterns of carbon emissions from irrigation", compromises the continuity of the text expressed in the paragraph below.
7 - The conclusion is well founded, together with the good quality of the English used in the text, which is clear and understandable. Congratulations.
Author Response
Reviewer #3
Reviewer #3: After detailed readings in the manuscript, entitled: "Research on Agricultural Cultivation Structure in Arid Areas Based on Water-Carbon nexus _ taking the middle reaches of the Heihe River as an example", the need for studies on food security and sustainable agricultural production is understood. The results of this research can effectively guide the future grain planting pattern towards water-saving and low-carbon agriculture in the arid zone. I suggest ACCEPT the manuscript with minor corrections.
Response:
Thank you very much for your recognition of the article! We are encouraged by your approval of our articles. Your comments help to improve the academic rigor of our articles.
The italic and underlined paragraphs are comments made by reviewers
The paragraphs with green colors are unchanged paragraphs in the original text.
The paragraphs with red colors are revised or added paragraphs in the original text.
Point 1: I suggest modifying the title: "Research on Agricultural Cultivation Structure in Arid Areas Based on Water-Carbon nexus _ taking the middle reaches of the Heihe River as an example", for "Agricultural Cultivation Structure in Arid Areas Based on Water-Carbon nexus _ taking the middle reaches of the Heihe River as an example".
Response 1: Thank you for your suggestion. This is a very kind offer. We have followed your suggestion and changed the title of the article to “Agricultural Cultivation Structure in Arid Areas Based on Water-Carbon nexus taking the middle reaches of the Heihe River as an example”.
Point 2: At the end of the Abstract, it is necessary to address the importance and need for this study on a global scale.
Response 2: Thank you for your suggestion. Due to the word limit of the summary section, we could not do a massive rewrite of it. However, we added a global dimension to it to interest the reader
The results of the research hold immense importance in guiding the future grain crop planting patterns for water-saving agriculture and low-carbon agriculture development in arid zones worldwide, aligning with the United Nations' Sustainable Development Goals. The results of the research can effectively guide the future grain crop planting pattern for water-saving agriculture and low-carbon agriculture development in global arid zone.
Point 3: The introduction is very well reasoned, but I suggest adding the importance of this study to the world at the end of the introduction. This would arouse the interest of readers more.
Response 3: Thank you for your suggestion. We present the implications of our study in terms of research implications for implementation in a global context. Our modifications are as follows:
Therefore, this research aims to establish the spatial distribution data of major food crops, namely maize and wheat, in the middle reaches of the Heihe River with town-level accuracy. This is accomplished by employing the Water-Carbon Nexus framework in a typical arid zone. In this paper, town-level data on irrigation water consumption and agricultural carbon emissions are obtained through farmer surveys, meteorological data, and satellite data. These data are then utilized to calculate the carbon emission intensity per unit of water consumption, thereby enabling the evaluation of the carbon emission capacity of crops. Subsequently, a comprehensive evaluation was conducted, which integrated considerations of food security, farm household income, and agroecological values. In the current global context, there is a pressing need for concerted efforts to mitigate the adverse impact of agriculture on the climate and ensure food production. Under the guidance of the United Nations Sustainable Development Goals (SDGs), the pursuit of a secure water environment, low-carbon emission environmental protection, sustainable agricultural development, and food security are globally unified objectives. The findings offer valuable guidance for the cultivation of food crops in arid regions worldwide, The research findings can serve as guidance for the cultivation of food crops in arid areas facilitate the exploration of spatial distribution patterns of water consumption and carbon emissions among different crops, and offer valuable insights towards achieving an optimal crop cultivation pattern that ensures food security.
Point 4: In Figure 1 can insert the coordinate grid. allowing other researchers to have the exact location of the study area.
Response 4: Thank you for your suggestion. This is very practical and valuable advice. We have modified Figure 1.
(Before)Figure 1. Zhangye City Location and Land Use Data
(After)Figure 1. Zhangye City Location and Land Use Data
Point 5: The methodology is well founded. The authors really did a good job. Congratulations.
Response 5: Thank you very much for your encouragement! Our team is very delighted to receive such feedback.
Point 6: I suggest deleting the subtitle: "3.4. Differences in spatial patterns of carbon emissions from irrigation", compromises the continuity of the text expressed in the paragraph below.
Response 6: Thank you for your suggestion. The title of this paragraph is very wrong. We have changed the title from “Differences in spatial patterns of carbon emissions from irrigation” to “Agricultural planting structure adjustment under linear programming model”
Point 7: The conclusion is well founded, together with the good quality of the English used in the text, which is clear and understandable. Congratulations.
Response 7: We are very pleased with your recognition of our work. With your support, I believe our team can achieve good results on this topic. Thank you very much!

Reviewer 4 Report
In the paper, the authors studied the irrigation water use efficiency and carbon emission differences of crops in arid areas and their spatial distribution using wheat and maize, two major food crops in the middle reaches of the Heihe River, as examples. This manuscript is well-designed and prepared. I would recommend the publication after a minor revision.
1. The authors should add more arguments to clarify the novelty of this manuscript. For example, the strength of the manuscript is that the authors combined different data sources and modeling analysis methods, however, this point was not emphasized properly since the reader has no understanding of the status of research methods for such a similar study.
2. What is DEM? Can you specify the full name?
3. What is a linear programming model? Can you explain this model specifically?
4. Please add more details on the software the author used (i.e. developer/manufacturer), also please clarify why you choose the software or model mentioned in the paper.
5. In line 248, please fixe the Chinese punctuation
6. “According to the results, the irrigation water consumption in the main basin of the Heihe River is significantly higher than that in the tributary basins, with Xintian Township (541 mm t-1), Shunhua Township (552 mm t-1), and Yonggu Township (600.56 321mm t-1) reporting particularly high levels of consumption. Interestingly, these towns also exhibit higher efficiency in terms of maize irrigation water use…” Please clarify which figure in the paper supports those statements.
7. “Accordingly, it can be inferred that wheat should be planted more in the lower reaches of the Heihe River and other areas where surface water is not sufficient from the perspective of planting structure because of its low corresponding capacity to surface water. This finding is consistent with the existing research results.” Can you add some citations to support the comparison between your study and the existing research results?
8. The authors mentioned the survey data. However, the results and discussion about this part were not presented. Please clarify what does the survey data use for.
The quality of the English language for the manuscript is fine, but please notice fix the Chinese punctuation
Author Response
Reviewer #4
Reviewer #4: In the paper, the authors studied the irrigation water use efficiency and carbon emission differences of crops in arid areas and their spatial distribution using wheat and maize, two major food crops in the middle reaches of the Heihe River, as examples. This manuscript is well-designed and prepared. I would recommend the publication after a minor revision.
Response:
Thank you very much for your recognition of the article! Your opinions helped to improve academic rigor of our article.
The italic and underlined paragraphs are comments made by reviewers
The paragraphs with green colors are unchanged paragraphs in the original text.
The paragraphs with red colors are revised or added paragraphs in the original text.
Point 1: The authors should add more arguments to clarify the novelty of this manuscript. For example, the strength of the manuscript is that the authors combined different data sources and modeling analysis methods, however, this point was not emphasized properly since the reader has no understanding of the status of research methods for such a similar study.
Response 1: Thank you for your suggestion. In the introductory section of this paper, we have added an interpretation of the current study, while in the last paragraph of the introductory section, we focus on the extension of the study to the current study. Examples of the changes are as follows:
In summary, the current research on agricultural planting structure primarily focuses on a national or regional level. In order to protect the interests of farmers and provide practical guidance for agricultural production, there is a need to construct a higher-precision model that can accurately reflect the spatial heterogeneity of water resource consumption and carbon emissions during irrigation processes. Furthermore, in order to better guide low-carbon agricultural production, particularly the adjustment of grain production scale, it is necessary to optimize the agricultural planting structure.
At the same time, the source of its data in general various studies are obtained from satellite data for a particular crop sum, lacking the exploration of the pattern of geographical distribution of crops. To guide agricultural cropping patterns more precisely and directly, it is particularly important to construct an irrigation model with higher accuracy and more specific to the production life of farmers.
This evaluation aimed to determine the optimal model for regional crop cultivation using a linear programming approach. The purpose of this research is to assess the ecological value of two crops by measuring the agricultural irrigation water consumption and carbon emission data of major food crops in a typical arid zone. And to determine the best ecological model for regional crop cultivation under the premise of ensuring food security using linear programming method. In the current global context, there is a pressing need for concerted efforts to mitigate the adverse impact of agriculture on the climate and ensure food production. The findings offer valuable guidance for the cultivation of food crops in arid regions worldwide, facilitate the exploration of spatial distribution patterns of water consumption and carbon emissions among different crops, and offer valuable insights towards achieving an optimal crop cultivation pattern that ensures food security.
Point 2: What is DEM? Can you specify the full name?
Response 2: Thank you for your suggestion. Our additions to this proper noun are as follows:
In this study, the planting distribution and yield distribution of two crops in the middle reaches of the Heihe River were determined using Landsat and MOD13 satellite data, along with Digital Terrain Model (DEM) elevation data and land use data provided by the China Geospatial Data Cloud.
Point 3: What is a linear programming model? Can you explain this model specifically?
Response 3: Thank you for your suggestion. We did not realize that our preconceived understanding of the model could lead to misunderstanding by the reader. While supplementing our interpretation of the model, we have added citations to it to support our use of the model. Our additions to the model are as follows:
A linear programming model is used to optimize the grain crop cultivation pat-tern in the middle reaches of the Heihe River. The linear programming problem is one of the most important branches of operations research, which studies the problem of maximizing (or minimizing) a linear objective under the constraints of a linear inequality or equation. The model is used extensively for planning agricultural cropping structures under multiple conditions [29-31]. In this research we can use this model to obtain the optimal solution for the acreage of two crops. The model contains objective functions and constraints. Among them, we mainly con-strain two crops at four levels: total carbon emission, irrigation water consumption, grain yield and farm household income. We use Linear Interactive and General Optimizer (LINGO) software to calculated.
- ACS S, BERENTSEN P B M, HUIRNE R B M. Conversion to organic arable farming in The Netherlands: A dynamic linear programming analysis [J]. Agricultural Systems, 2007, 94(2): 405-15, doi:https://doi.org/10.1016/j.agsy.2006.11.002.
- BAUER S, KASNAKOGLU H. Non-linear programming models for sector and policy analysis: Experiences with the Turkish agricultural sector model [J]. Economic Modelling, 1990, 7(3): 275-90, doi: https://doi.org/10.1016/0264-9993(90)90013-T.
- BARTOLINI F, BAZZANI G M, GALLERANI V, et al. The impact of water and agriculture policy scenarios on irrigated farming systems in Italy: An analysis based on farm level multi-attribute linear programming models [J]. Agricultural Systems, 2007, 93(1): 90-114, doi: https://doi.org/10.1016/j.agsy.2006.04.006.
Point 4: Please add more details on the software the author used (i.e. developer/manufacturer), also please clarify why you choose the software or model mentioned in the paper.
Response 4: Thank you for your suggestion. We have further described for all software used in the article:
To begin with, Landsat 8 data was employed to obtain the Normalized Difference Vegetation Index (NDVI) for the region. Through radiometric correction, atmospheric correction, and image cropping using the ENVI software, the precise NDVI temporal curve of the region was derived. ENVI software is developed by Exelis Visual Information Solutions, the main purpose of this software is to process and analyze remote sensing data.
The second step entails employing ArcGIS to establish a data model for regional energy consumption, utilizing the spatial distribution of planting areas as a foundation. ArcGIS software was developed by Esri (Environmental Systems Research Institute) to manage, analyze and visualize geographic data.
The linear programming problem is one of the most important branches of operations research, which studies the problem of maximizing (or minimizing) a linear objective under the constraints of a linear inequality or equation. The model is used extensively for planning agricultural cropping structures under multiple conditions.
Point 5: In line 248, please fixe the Chinese punctuation
Response 5: We are very sorry for such a low-level mistake. We have corrected this.
Point 6: “According to the results, the irrigation water consumption in the main basin of the Heihe River is significantly higher than that in the tributary basins, with Xintian Township (541 mm t-1), Shunhua Township (552 mm t-1), and Yonggu Township (600.56 321mm t-1) reporting particularly high levels of consumption. Interestingly, these towns also exhibit higher efficiency in terms of maize irrigation water use…” Please clarify which figure in the paper supports those statements.
Response6: Thank you for your suggestion. Our additions to this part of the results are as follows:
In the case of wheat, the irrigation water efficiency exhibits a distinctly opposite trend compared to maize. Wheat production in the middle reaches of the Heihe River is concentrated in the upper basin of the river. Spatially speaking, this region can be roughly divided into two areas: the main basin of the Heihe River, which encompasses towns in the southwest of the planting area, and the tributary basins, which include towns in the northeast. According to the results, the irrigation water consumption in the main basin of the Heihe River is significantly higher than that in the tributary basins, with Xintian Township (541 mm t-1), Shunhua Township (552 mm t-1), and Yonggu Township (600.56 mm t-1) reporting particularly high levels of consumption. Interestingly, these towns also exhibit higher efficiency in terms of maize irrigation water use. Interestingly, these towns also exhibit a higher level of water consumption for maize irrigation. Their irrigation water consumption per unit yield was higher in Xintian Town (728 mm t-1), Sunhua Town (551 mm t-1) and Yonggu Town (605 mm t-1) than the average water consumption of maize (515.6 mm t-1).
Point 7: “Accordingly, it can be inferred that wheat should be planted more in the lower reaches of the Heihe River and other areas where surface water is not sufficient from the perspective of planting structure because of its low corresponding capacity to surface water. This finding is consistent with the existing research results.” Can you add some citations to support the comparison between your study and the existing research results?
Response7: Thank you for your suggestion. We have added a new quote to this statement and enhanced the argument for the point in the following paragraph.
Accordingly, it can be inferred that wheat should be planted more in the lower reaches of the Heihe River and other areas where surface water is not sufficient from the perspective of planting structure because of its low corresponding capacity to surface water. This finding is consistent with the existing research results [48]. From the water footprint perspective, the green and blue water footprints of maize were 0.457 and 0.058 m3 kg-1, respectively, compared to 0.55 and 0.114 m3 kg-1 for wheat [49]. This reflects the fact that wheat is more responsive to both precipitation and surface water resources than maize. This implies that for water-scarce areas, the incremental demand for irrigation water will be much greater for wheat than for maize.
- ZHANG Y, LIU W, CAI Y, et al. Decoupling analysis of water use and economic development in arid region of China – Based on quantity and quality of water use [J]. Science of The Total Environment, 2021, 761: 143275, doi: https://doi.org/10.1016/j.scitotenv.2020.143275.
Point 8: The authors mentioned the survey data. However, the results and discussion about this part were not presented. Please clarify what does the survey data use for.
Response 8: Thank you for your question about this issue. Our use of survey data is mainly focused on the methods section in Chapter 2. In it, 2.2.4 details the process of obtaining carbon emission data for agricultural irrigation from survey data. Tables have been added to the methods section to better establish the link between crop cultivation in the survey data domain.
Survey data is mainly derived from individual interviews conducted with local farmers. In order to demonstrate the significant influence of topographic factors and the distribution of irrigation water resources on the irrigation process, this study specifically selected four representative villages in the middle reaches of the Heihe River Basin, where maize and wheat cultivation is predominant. Each farming household cultivates a non-separable area of 1-2 hectares for both crops. The proportional distribution of major cereal crop cultivation areas in each county is presented in Table 1. These villages are situated in di-verse geographical and topographical settings. The distribution of questionnaires took place in February 2023, resulting in a collection of 421 completed questionnaires. Among them, 409 questionnaires were deemed qualified, while 12 were considered unqualified, yielding a qualification rate of 97.1%. To account for the significant influence of terrain factors and the distribution of irrigation water resources on the irrigation process, this study specifically selects four representative villages situated in distinct geographical and terrain features within the primary food crop growing area of the middle reaches of the Heihe River basin. The survey questionnaire was distributed in February 2023, resulting in the collection of a total of 421 questionnaires. Among these, 409 questionnaires were deemed qualified, while 12 were considered unqualified, yielding a qualification rate of 97.1%.
Table 1. The proportional distribution of major cereal crop cultivation areas.
|
Point Number |
Wheat |
Corn |
Tubers |
|
Zhangye City (Total) |
0.17 |
0.48 |
0.10 |
|
Ganzhou District |
0.04 |
0.92 |
0.00 |
|
Sunan County |
0.14 |
0.27 |
0.03 |
|
Minle County |
0.38 |
0.13 |
0.31 |
|
Linze County |
0.01 |
0.97 |
0.00 |
|
Gaotai County |
0.10 |
0.65 |
0.04 |
|
Shandan County |
0.42 |
0.03 |
0.23 |

Round 2
Reviewer 1 Report
Please revised the whole paper according to the editor comments.
Author Response
Reviewer
Please address Reviewer 1 comments - the author need provide some references in the equations. Another, the author needs concise the conclusions section.
Response:
Thank you for your valuable suggestions and thorough review of the article. Your feedback greatly enhances the academic rigor of our work.
The italic and underlined paragraphs are comments made by reviewers
The paragraphs with green colors are unchanged paragraphs in the original text.
The paragraphs with red colors are revised or added paragraphs in the original text.
Point 1: The author need provide some references in the equations.
Response 1: Thank you for your suggestion. To improve the rigor of the article, we have added () citations to the article to increase the rigor of our use of formulas. Our changes are as follows:(Formula refer to the Annex)
Effective precipitation (EP) refers to the portion of precipitation that successfully infiltrates the soil and is absorbed and utilized by crops. Empirical models are commonly used to calculate EP based on meteorological precipitation data. In this study, the calculation model utilized is the United States Department of Agriculture Soil Conservation Service (USDA SCS) model [28]. The specific formula is as follows:
Where R represents daily precipitation.
The linear programming problem is one of the most important branches of operations research, which studies the problem of maximizing (or minimizing) a linear objective under the constraints of a linear inequality or equation. The model is used extensively for planning agricultural cropping structures under multiple conditions [29-31]. In this research we can use this model to obtain the optimal solution for the acreage of two crops. The model contains objective functions and constraints. Among them, we mainly constrain two crops at four levels: total carbon emission, irrigation water consumption, grain yield and farm household income. We use Linear Interactive and General Optimizer (LINGO) software to calculated [32]. The constraint model is as follows:
Where F(x) represents Carbon emissions for both crops; XM and XW are the crop area planted to maize and wheat, respectively, and CM and CW are the current average carbon emission factors per hectare planted to maize and wheat, respectively.
In order to minimize the irrigation water consumption for agricultural production without affecting the farmers’ profitability, we constructed three levels of constraints.
Constraint 1: Total planted area. This study uses the area planted to maize and wheat crops as independent variables. Considering the future urban expansion, the current total planted area is set to the maximum value:
Where represents the total planted area.
Constraint 2: Irrigation water security. Since the irrigation water consumption per planted area is different for the two crops, the total irrigation water consumption needs to be controlled when performing the irrigation pattern update [33,34]. The expressions are as follows:
Where represents the water consumption per unit area of crop cultivation; represents total current irrigation water consumption.
Constraint 3: Income security for farmers. When restructuring the regional irrigation system, the stability of farmers' income needs to be guaranteed and the increase in farmers' income is considered as a win-win strategy. Since the policy of Zhangye City encourages the planting of seeded maize and the proportion of seeded maize planting has increased year by year, this study brings in the income per unit area of maize planting with the price of seeded maize [35]. The expressions are as follows:
Where represents farmers' income per unit area of crop cultivation.
Constraint 4: Ensure food security. According to the data of Zhangye City Bureau of Statistics, both studied food crops can be exported to satisfy the region's own self-sufficiency. To ensure regional food security, the production of each grain should not be lower than the region's needs [36]. The expressions are as follows:
(14)
Where represents the Grain yield per unit area; represents total food production of both crops.
- ZAREI A R, MOGHIMI M M. Modified version for SPEI to evaluate and modeling the agricultural drought severity [J]. International journal of biometeorology, 2019, 63: 911-25, doi: https://doi.org/10.1007/s00484-019-01704-2.
- ACS S, BERENTSEN P B M, HUIRNE R B M. Conversion to organic arable farming in The Netherlands: A dynamic linear programming analysis [J]. Agricultural Systems, 2007, 94(2): 405-15, doi:https://doi.org/10.1016/j.agsy.2006.11.002.
- BAUER S, KASNAKOGLU H. Non-linear programming models for sector and policy analysis: Experiences with the Turkish agricultural sector model [J]. Economic Modelling, 1990, 7(3): 275-90, doi: https://doi.org/10.1016/0264-9993(90)90013-T.
- BARTOLINI F, BAZZANI G M, GALLERANI V, et al. The impact of water and agriculture policy scenarios on irrigated farming systems in Italy: An analysis based on farm level multi-attribute linear programming models [J]. Agricultural Systems, 2007, 93(1): 90-114, doi: https://doi.org/10.1016/j.agsy.2006.04.006.
- Xia, C.; Zhang, J.; Zhao, J.; Xue, F.; Li, Q.; Fang, K.; Shao, Z.; Zhang, J.; Li, S.; Zhou, J. Exploring potential of urban land-use management on carbon emissions—— A case of Hangzhou, China. Ecological Indicators 2023, 146, 109902,139-144, doi:https://doi.org/10.1016/j.ecolind.2023.109902.
- Karandish, F.; Šimůnek, J. A comparison of the HYDRUS (2D/3D) and SALTMED models to investigate the influence of various water-saving irrigation strategies on the maize water footprint. Agricultural Water Management 2019, 213, 809-820, doi: https://doi.org/10.1016/j.scitotenv.2019.135587.
- Yang, G.; Tian, L.; Li, X.; He, X.; Gao, Y.; Li, F.; Xue, L.; Li, P. Numerical assessment of the effect of water-saving irrigation on the water cycle at the Manas River Basin oasis, China. Science of The Total Environment 2020, 707, 135587, doi: https://doi.org/10.1016/j.agwat.2018.11.023.
- Xiao, X.; Fan, L.; Li, X.; Tan, M.; Jiang, T.; Zheng, L.; Jiang, F. Water-use efficiency of crops in the arid area of the middle reaches of the Heihe River: taking Zhangye City as an example. Water 2019, 11, 1541, 151-166, doi: https://doi.org/10.3390/w11081541.
- Li, W.; Ruiz-Menjivar, J.; Zhang, L.; Zhang, J. Climate change perceptions and the adoption of low-carbon agricultural technologies: Evidence from rice production systems in the Yangtze River Basin. Science of The Total Environment 2021, 759, 143554, doi:https://doi.org/10.1016/j.scitotenv.2020.143554.
Point 2: the author needs concise the conclusions section.
Response 2: Thank you for your suggestion. We have extensively revised the conclusions section of the paper, aiming to succinctly and clearly express our ideas. After we streamlined the main part of the conclusion, the word count of the conclusion section was changed from 375 words to 237 words. Our changes are as follows:
Within the framework of low-carbon agriculture and water-saving agriculture, an examination of the cropping structure in arid regions holds potential for the optimization of forthcoming agricultural practices. By specifically examining the primary food crops cultivated in the middle reaches of the Heihe River, this study establishes precise county-level water-carbon relationships. It further investigates alterations in the spatial distribution of irrigation water utilization and carbon emissions for these two crops. Additionally, a linear programming model is employed to formulate a low-carbon agricultural cropping structure, taking into account factors such as food security and farmers' income. This study directs its attention to the principal food crops, namely maize and wheat, cultivated in the middle reaches of the Heihe River. An assessment of the food crop-ping structure in this arid locale is conducted by means of evaluating the spatial con-figuration of its Water-carbon nexus. To acquire the findings of this study, a three-step approach is adopted. Initially, irrigation water consumption data is derived through the utilization of meteorological data, specifically precipitation data and crop evapotranspiration data. Subsequently, empirical data pertaining to irrigation processes, including water diversion and the engagement of water extraction machinery, is obtained through field visits to farmers. This empirical data is further supplemented by measurements of energy consumption. Finally, carbon emission data is obtained em-ploying the IPCC coefficient method. Our conclusions are as follows: (1) The carbon emissions per unit of water consumption for maize are 0.03×10-6t mm-1, and 0.49×10-6t mm-1for wheat. The carbon emissions per unit of irrigation water consumed by maize cultivation are lower than those of wheat. (2) The main grain crops in the middle reaches of the Heihe River basin are maize and wheat. From the perspective of water saving, spring wheat cultivation exhibits more water saving ability in the arid zone. However, the carbon emission is 18.58t t-1, while for wheat, the carbon emission is 20.88t t-1. On average, the carbon emission of maize is lower than that of wheat cultivation. In terms of spatial patterns. In areas with relatively abundant surface water in the upper reaches of the Heihe River, the irrigation water consumption per unit of production is lower for maize, while the irrigation water consumption is higher for wheat. The spatial distribution of irrigation water allocation for maize and wheat revealed distinctive patterns. Maize cultivation displayed lower demand for irrigation water in the growing areas near the upper reaches of the Heihe River, while higher demand was observed in the lower reaches. Furthermore, carbon emissions from both crops were relatively low in the upstream regions. (3) After taking into account food security and energy conservation, the optimal cropping structure is 136203.9 ha of maize and 21019.93 ha of wheat. The area planted with maize can be appropriately increased, while the area planted with wheat can be appropriately reduced under the premise of food security. As a result, the cultivated area for maize expands to 104% of its original size, while the area for wheat reduces to 59% of its original size. These findings demonstrate that growing maize in areas with relatively abundant surface water for arid regions can help reduce carbon emissions from irrigated agriculture, while wheat cultivation is more suitable for areas with insufficient surface water.
